# A tradeoff between acoustic and linguistic feature encoding in spoken language comprehension

**Filiz Tezcan[1]\*, Hugo Weissbart[2], Andrea E Martin[1,2]**

[1]Language and Computation in Neural Systems Group, Max Planck Institute for Psycholinguistics, Nijmegen, Netherlands; [2]Donders Centre for Cognitive Neuroimaging, Radboud University, Nijmegen, Netherlands

**\*For correspondence:**
Filiz.TezcanSemerci@mpi.nl

**Abstract** When we comprehend language from speech, the phase of the neural response aligns with particular features of the speech input, resulting in a phenomenon referred to as *neural tracking*. In recent years, a large body of work has demonstrated the tracking of the acoustic envelope and abstract linguistic units at the phoneme and word levels, and beyond. However, the degree to which speech tracking is driven by acoustic edges of the signal, or by internally-generated linguistic units, or by the interplay of both, remains contentious. In this study, we used naturalistic story-listening to investigate (1) whether phoneme-level features are tracked over and above acoustic edges, (2) whether word entropy, which can reflect sentence- and discourse-level constraints, impacted the encoding of acoustic and phoneme-level features, and (3) whether the tracking of acoustic edges was enhanced or suppressed during comprehension of a first language (Dutch) compared to a statistically familiar but uncomprehended language (French). We first show that encoding models with phoneme-level linguistic features, in addition to acoustic features, uncovered an increased neural tracking response; this signal was further amplified in a comprehended language, putatively reflecting the transformation of acoustic features into internally generated phoneme-level representations. Phonemes were tracked more strongly in a comprehended language, suggesting that language comprehension functions as a neural filter over acoustic edges of the speech signal as it transforms sensory signals into abstract linguistic units. We then show that word entropy enhances neural tracking of both acoustic and phonemic features when sentence- and discourse-context are less constraining. When language was not comprehended, acoustic features, but not phonemic ones, were more strongly modulated, but in contrast, when a native language is comprehended, phoneme features are more strongly modulated. Taken together, our findings highlight the flexible modulation of acoustic, and phonemic features by sentence and discourse-level constraint in language comprehension, and document the neural transformation from speech perception to language comprehension, consistent with an account of language processing as a neural filter from sensory to abstract representations.

## Editor's evaluation

This study addresses a fundamental aspect of human speech processing: namely, how acoustic and linguistic features interact during comprehension. The authors present convincing evidence that helps elucidate the role of language experience on neural processing, re-weighting processing of speech based on whether a listener understands the language being spoken.

## Introduction

When we understand spoken language, we transform a continuous physical signal into structured meaning. In order to achieve this, we likely capitalize on stored, previously-learned linguistic representations and other forms of knowledge. During this process, systematic changes shown by the acoustic signal are categorized by the brain as phonemes, while the similar ordering of consecutive phonemes creates words and, a certain ordering of words creates phrases and sentences. As these transformation occur, the phase of the neural signal is thought to align with temporal landmarks in both acoustic and linguistic dimensions; this phenomenon is referred to as neural tracking (*Daube et al., 2019*; *Luo and Poeppel, 2007*; *Di Liberto et al., 2015*; *Brodbeck et al., 2018*; *Broderick et al., 2018*; *Daube et al., 2019*; *Keitel et al., 2018*; *Donhauser and Baillet, 2020*; *Gillis et al., 2021*; *Kaufeld et al., 2020a*; *Weissbart et al., 2020*; *Brodbeck et al., 2022*; *Heilbron et al., 2020*; *Coopmans et al., 2022*; *Slaats et al., 2023*; *Ten Oever et al., 2022a*; *Zioga et al., 2023*). Neural tracking has been proposed as a mechanism to segment the acoustic input into linguistic units such as phoneme, syllables, morphemes (*Giraud and Poeppel, 2012*; *Ghitza, 2013*; *Ding et al., 2016*) by aligning the neural excitability cycles to the incoming input. One way to quantify the degree of neural tracking of speech features is using a linear encoding modelling also known as Temporal Response Functions (TRF). Adding sub-lexical, lexical, and phrase- and sentence-level predictors, either quantified with information theoretic metrics (e.g. surprisal and entropy), or syntactic annotations, to the encoding model improves the reconstruction accuracy of neural responses on distinct sources in the brain (see *Brodbeck et al., 2022*), and surprisal and entropy predictors have been shown to modulate the neural response at different time intervals (see *Donhauser and Baillet, 2020*). However, whether acoustic features and higher level linguistic units are encoded with similar veracity during comprehension, and how they impact each other's encoding is still poorly understood. In this study, we investigated the effect of language comprehension in a first language (Dutch) on the tracking of acoustic and phonemic features, contrasted with a statistically familiar but uncomprehended language (French), during naturalistic story listening. Contrasting a comprehended language with a language that is familiar, but not understood, allows us to separate neural signals related to speech processing vs. those related to speech processing in the service of language comprehension. We asked three questions: (1) whether phoneme-level features contribute to neural encoding even when acoustic contributions are carefully controlled, as a function of language comprehension, (2) whether sentence- and discourse-level constraints on lexical information operationalized as word entropy impacted the encoding of acoustic and phoneme-level features, and (3) whether tracking of acoustic landmarks (viz., acoustic edges) was enhanced or suppressed as a function of comprehension. We found that acoustic features are enhanced when the spoken language stimulus is not understood, but that phonemic features are more strongly encoded when language is understood, consistent with an account where gain modulation, in the form of enhancement or suppression, expresses the behavioral goal of the brain (*Martin, 2016*; *Martin, 2020*). We found that both phonemic and acoustic features are more strongly encoded when the context is less constraining, as exemplified by word entropy. Finally, and in contrast with extant arguments in the literature about the pre-activation or enhancement of low-level representations with the contextual information (*DeLong et al., 2005*; *McClelland and Rumelhart, 1981 Rumelhart and McClelland, 1982*, *Nieuwland et al., 2018*; *Nieuwland, 2019* ), we found that acoustic-edge processing appears to be suppressed when language comprehension occurs, compared to when it does not, and both acoustic and phonemic features are suppressed when entropy is low within a comprehended language (viz., the context is more constraining and a word is more expected). This pattern of results is consistent with acoustic edges being important for auditory and speech processing, but speaks against their functioning as computational landmarks during language comprehension, the behavioral and neural goal of speech processing.

During comprehension, contextual information from higher level linguistic units such as words, phrases, and sentences likely affects the neural representation and processing of lower level units like phonemes and acoustic features. A wealth of behavioral and neural data from both spoken and visual language comprehension contexts has been leveraged to illustrate the incremental processing of incoming perceptual input within existing sentence and discourse context. During spoken language comprehension, acoustic, phonemic, phonological, and prosodic information is dynamically integrated with morphemic, lexical, semantic, and syntactic information (e.g., *Bai et al., 2022*; *Kaufeld et al., 2020b*; *Kaufeld et al., 2020c*; *Marslen-Wilson and Welsh, 1978*; *Friederici, 2002*; *Hagoort,*

*2013*; *Martin et al., 2017*; *Oganian et al., 2023*; *Martin, 2016*; *Martin, 2020*). Models with top-down and bottom-up information flow have been invoked to account for how sensory information could be integrated with internally-generated knowledge or information, expressed in many forms in different models, as statistical priors, distributional knowledge, or as abstract structure (*McClelland and Rumelhart, 1981*; *Rumelhart and McClelland, 1982*; *Rao and Ballard, 1999*; *Lee and Mumford, 2003*; *Friston, 2005*; *Martin, 2016*; *Martin and Doumas, 2019*; *Martin, 2020*; *Ten Oever and Martin, 2021*; *Ten Oever et al., 2022a*; *Ten Oever et al., 2022b*). In word recognition, models such as the Interactive Activation Model (*McClelland and Rumelhart, 1981*) and TRACE (*Rumelhart and McClelland, 1982*), feedback connections from word-level to phoneme-level enable a faster activation of phonemes in words than phonemes in nonwords by either enhancing the representation of expected phonemes or suppressing the competing phonemes. Even though these models can account for phenomena such as the word-superiority effect, faster detection of a letter on a masked visual image when presented in a word than in an unpronounceable nonword or in isolation (*Reicher, 1969*; *Wheeler, 1970*; *Mewhort, 1967*), they do not incorporate effects of sentence- and discourse- level constraints. It has also been demonstrated that mispronounced or ambiguous phonemes are more likely to be missed by participants during a detection task when the context is more constraining, indicating that top-down processing constraints interact with bottom-up sensory information to reduce the number of possible word candidates as the acoustic input unfolds (*Marslen-Wilson and Welsh, 1978*; *Martin et al., 2017*; *Martin and Doumas, 2017*).

Even though the effects of sentence-level and word-level context on sub-lexical representations were widely studied with behavioral experiments, studies that investigated the neural readout of the perceptual modulation by the context are still very limited for non-degraded or naturally produced spoken stimuli. *Martin, 2016*; *Martin, 2020* suggested that cues from each hierarchical level are weighted according to their reliabilities and integrated simultaneously by gain modulation in the form of selective amplification and inhibition. When the acoustic edges of the speech signal reach a threshold for the activation of a particular phoneme representation, the sensory representations of the acoustic edges are inhibited. On a slower timescale, the resulting representations of phonemes activate items in the lexicon which then, in turn, suppress the phonemic representations once lexical access is achieved. Intelligibility of the speech signal determines the reliability of the acoustic signal, and sentence- and discourse-level constraint determines the reliability of lexical access and structure formation. According to this theoretical model, when context is less constraining, sub-lexical representations are less inhibited compared to in a highly constraining context.

Linear encoding models have made it possible to investigate how certain features of phonemes and words affect the neural response during naturalistic language comprehension. However, one of the important challenges in this modeling method is the difficulty of understanding whether the features tracked by the brain are only acoustic changes or changes in linguistic units as well (*Daube et al., 2019*). For example, the beginning of a word may coincide with the beginning of a phoneme, that is also a sudden change in the envelope of the acoustic signal. It has also been argued that neural tracking reflects the convolution of evoked responses to the change in acoustic envelope referred to as acoustic edges (*Oganian et al., 2023*). To overcome this, features that can cause changes in brain signal should be modeled together as multivariate and to isolate the effect of each feature, features should be added to the models in an incremental way (*Brodbeck et al., 2021*). Most of the studies using the linear encoding models showed the tracking of either the envelope of the acoustic signal or linguistic units at both word and phoneme levels however only a few of them controlled the acoustic features. In this study, we added each phoneme feature on top of the acoustic base model incrementally and compared the reconstruction accuracies only explained by that feature.

Another limitation of linear encoding method is that only the linear relationship between the changes in the features and the amplitude of the brain signal can be observed. While modeling the linear relationship of the conditional probability of phonemes in words with the brain signal, the effect of phoneme features on the brain signal is assumed to be temporally and topologically constant. However, previous studies have shown that the intensity of the sound stimulus, and the constraints introduced on the words by the context of the sentence has an effect on the latency of the brain signal. Although it is not possible to continuously model this dynamic relationship with linear models, stimulus features can be separated according to certain properties and modeled separately (*Drennan and Lalor, 2019*). Thus, it is possible to model how sentence and discourse context affects

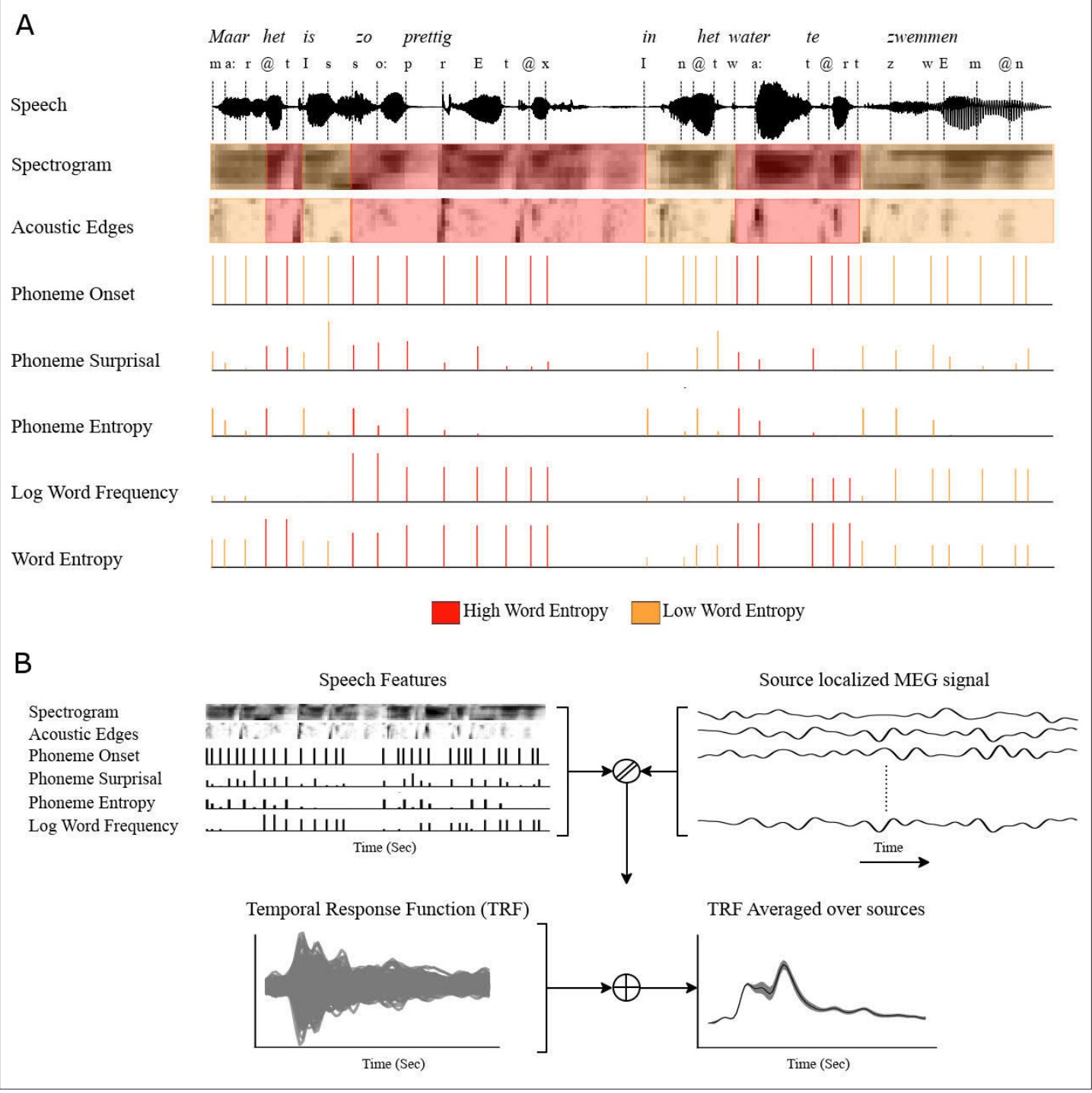

**Figure 1.** Schematic of TRF models and features used in models. (**A**) All speech features are divided into high and low word entropy conditions (**B**) TRFs for each brain source are generated by the linear regression model that estimates the source localized MEG signal from speech features, then they are averaged over sources.

the tracking over lower-level features by separating the predictors into high and low constraining context conditions, denoted by differences in word entropy. *Figure 1* shows predictors are separated into low and high word entropy conditions.

To this end, we made use of an MEG dataset of native Dutch speakers with no or minimal comprehension of French (i.e. self-report of no comprehension,<1 year education in French, chance performance on comprehension questions) that was recorded while they listened to audiobooks in Dutch

and French. We then used the linear encoding modeling method to predict the MEG signal from the acoustic and phoneme-level speech features including the spectrogram, acoustic edges, phoneme onset, phoneme surprisal and phoneme entropy, separately, for words with high and low word entropy. Word entropy was used to quantify the sentence- and discourse-level constraint for each word given the preceding 30 words. Phoneme surprisal quantifies the conditional probability of each phoneme in a word calculated according to the probability distribution over the lexicon weighted by the occurrence frequency of each word. Phoneme entropy reflects how constraining each phoneme is by quantifying the uncertainty about the next phoneme with each unfolding phoneme. If sentence comprehension modulates phoneme-level processing then the reconstruction accuracies should differ in two important contrasts. First, there should be an effect of comprehended language on whether words with high and low entropy affect phoneme processing, and within the comprehended language, high and low entropy words should differently affect phoneme features. For example, when participants are listening to an audiobook in a language they can understand, they can encounter a word like 'interval' or 'internet' in a high or low constraining sentence. The occurrence frequency of the word 'internet' is higher than 'interval' and the conditional probability of hearing the phoneme 'n' after 'ɪnˈtɜː(r)' is also higher than phoneme 'v'. That means the surprisal of phoneme 'n' which is calculated by the negative log probability, is lower. Representations for phonemes might be different when the reliabilities of contextual cues are different. In a sentence such as 'You can find anything on the internet' phoneme features might be more important and more informative compared to a sentence like 'On campus, you can connect to wireless internet' because in the first sentence it is more difficult to predict which word is coming given the previous words before the word 'internet' in the sentence, so participants may rely more on the phoneme-level features.

Our aim in this current study was to investigate the following questions: (1) Can we measure neural tracking of phoneme- and word-level features when acoustic features are properly controlled? (2) Can we demonstrate that products of language comprehension (viz., words in a sentence and story context) modulate the encoding of lower-level linguistic cues (acoustic and phoneme level features) under normal listening conditions? And (3) does the encoding of acoustic information change as a function of comprehension, reflecting perceptual modulation of neural encoding in the service of behavioral goals? In order to address these questions, we chose to compare the neural response between a comprehended first language (Dutch) and an uncomprehended but acoustically familiar one (French). Having a high-familiarity, statistically known control language allows us to model the neural response to acoustic and phoneme-level features and contrast it to a case where there is comprehension versus where there is not. This is stronger control than examining the neural response to an unknown, statistically unfamiliar language, whose modulation of the neural response may mask the degree to which acoustic and phoneme-level processing dominate the neural signal during language comprehension.

## Results

### Behavioral results

To evaluate if the stories were comprehended by participants, we compared the percentage of correct answers to comprehension questions after each story part. Participants replied 88% (SD = 7%) of questions, significantly above the chance level 25% (t(46)=39.45, p<0.0001) about Dutch stories correctly and 25% (SD = 11%) of questions about French stories which was not significantly different than the chance level (t(46)=0.44, p=0.66).

### Tracking performance of linguistic features

Firstly, to investigate the question whether linguistic units are tracked by the brain signal even when acoustic features are controlled, we compared the averaged reconstruction accuracies over all source points to see if adding each feature is increasing the reconstruction accuracy. To assess the individual contribution of each feature toward reconstruction accuracy, we conducted a stepwise analysis by fitting various TRF models. We added each feature sequentially to an acoustic base model, and then calculated the difference in accuracy between the model with the feature of interest and the previous model without that feature. Detailed information can be found in Table 11 in the Materials and methods section. We then fitted a linear mixed model for reconstruction accuracies with random intercept for subjects and a random slope for fixed effects. Independent variables were language

**Table 1.** LMM results of reconstruction accuracies for Dutch and French stories.

|  | Estimate | Std. Error | t value | Pr(>|t|) |  |
|---|---|---|---|---|---|
| (Intercept) | 2.86E-03 | 2.38E-04 | 12.00 | 2.16E-11 | *** |
| Language (French - Dutch) | −7.70E-04 | 1.57E-04 | −4.91 | 5.64E-05 | *** |
| Phon. Onset – Acoustic | 5.50E-05 | 1.14E-05 | 4.82 | 3.04E-06 | *** |
| Phon. Surprisal – Phon. Onset | 7.90E-05 | 1.14E-05 | 6.91 | 7.51E-11 | *** |
| Phon. Entropy – Phon. Surprisal | 1.33E-04 | 1.14E-05 | 11.63 | 2.00E-16 | *** |
| Word Frequency – Phon. Entropy | 1.48E-04 | 1.14E-05 | 12.95 | 2.00E-16 | *** |
| Language: Phon. Onset – Acoustic | −3.64E-05 | 1.62E-05 | −2.25 | 2.55E-02 | * |
| Language: Phon. Surprisal – Phon. Onset | −5.97E-05 | 1.62E-05 | −3.70 | 2.88E-04 | *** |
| Language: Phon. Entropy – Phon. Surprisal | −1.04E-04 | 1.62E-05 | −6.45 | 9.83E-10 | *** |
| Language:Word Frequency – Phon. Entropy | −1.30E-04 | 1.62E-05 | −8.07 | 9.12E-14 | *** |

****<0.0001, ***<0.001, **<0.01, *<0.05.

(Dutch and French) and models (Acoustic, Phoneme Onsets, Phoneme Surprisal, Phoneme Entropy and, Word Frequency). Linear mixed model (LMM) with a random slope both for models and language did not converge as the reconstruction accuracies were highly correlated, so we only fitted a random slope for language. We used backward difference coding for the model contrasts, so each contrast shows the difference between consecutive models (e.g. In *Table 1*, Phoneme Onset row shows the contrast between the accuracies of Phoneme Onset model and Acoustic model.) To evaluate weather adding language and models and their interaction as fixed effect increased predictive accuracy, we compared LMMs with and without these effects using R's anova() function.

The formulas used for the LMMs were then:

$$\mathrm{LMM_1 : Accuracy \sim Language * Models + (1 + Language|subject)}$$
$$\mathrm{LMM_2 : Accuracy \sim Language + Models + (1 + Language|subject)}$$
$$\mathrm{LMM_3 : Accuracy \sim Models + (1|subject)}$$
$$\mathrm{LMM_4 : Accuracy \sim 1 + (1|subject)}$$

LMM comparison showed that Models (LMM3 - LMM4 $\Delta\chi^2$ = 82.79, *P*<0.0001, LLM3 Bayesian Information Criterion (BIC): –3633.1, LMM4 BIC: –2787.5), Language (LMMl2 – LMM3 $\Delta\chi^2$ = 862.02, *P*<0.0001, LLM2 BIC: –3693.9), and their interaction (LMM1 - LMM2 $\Delta\chi^2$ = 71.63, p<0.0001, LLM1 BIC: –3743.7) predicted the averaged reconstruction accuracies. We reported the results of LMM1 in *Table 1* as it was the LMM with most predictive power and lowest Bayesian Information Criterion (BIC).

Averaged reconstruction accuracies were significantly higher in Dutch stories. Each feature incrementally increased averaged reconstruction accuracy compared to previous model and there was a significant interaction between Language and Models (*Table 1*).

Then we run two separate mixed-effect models for French and Dutch stories with random intercept for subjects and a random slope for models. Each feature incrementally increased averaged reconstruction accuracy compared to previous model for each language (*Figure 2A*, *Figure 2B*, *Table 2* and *Table 3*).

Then, we identified the source points that each feature changed the reconstruction accuracy compared to previous model using a mass-univariate two tail related sample t-test with threshold-free cluster enhancement (TFCE). *Figure 2C* shows the sources where reconstruction accuracies of base acoustic model were significantly different than zero. For both languages, acoustic features were tracked on both hemispheres around language network. *Figure 2-D* shows the sources where each feature incrementally increased the reconstruction accuracy compared to previous model. We fitted a liner mixed effect model to test the accuracy improvement by each linguistic feature for lateralization by taking the average of the contrasts shown in *Figure 2-D* for each hemisphere. We used the below formulas for LMM.

$$\mathrm{LMM_1 : Accuracy \sim Hemisphere + (1|subject)}$$

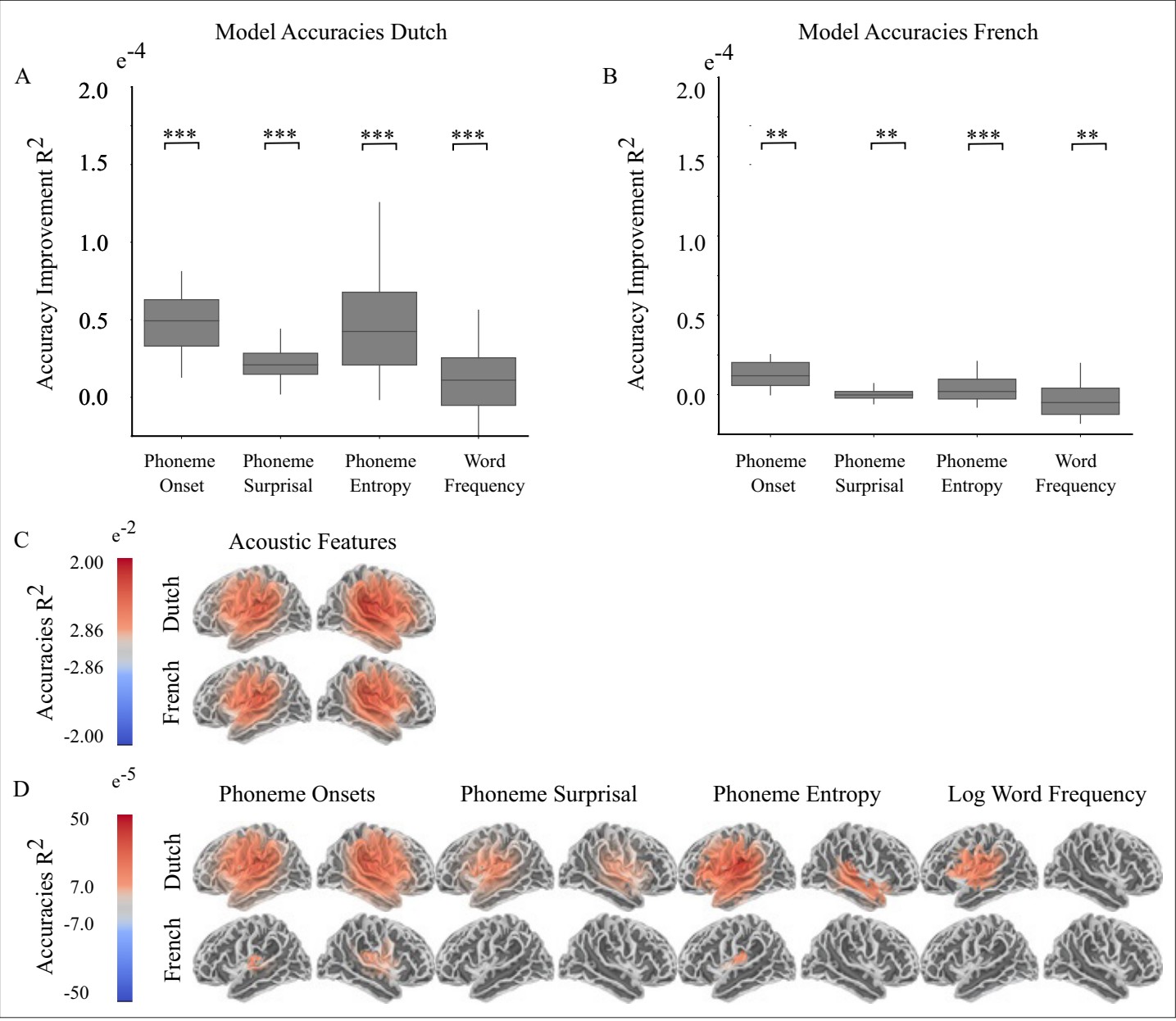

**Figure 2.** Model accuracy comparison between Dutch and French stories (n=24). (**A**) Accuracy improvement (averaged over the sources in whole brain) by each feature for Dutch Stories (**B**) Accuracy improvement (averaged over the sources in whole brain) by each feature for French Stories. Braces in Figure A and B shows the significance values of the contrasts (difference between consecutive models, ****<0.0001, ***<0.001, **<0.01, *<0.05) in linear mixed effect models (*Tables 2 and 3*). Error bars show within subject standard errors. (**C**) Source points where accuracies of base acoustic model were significantly different than 0 (**D**) Source points where reconstruction accuracies of the model were significantly different than previous model. Accuracy values shows how much each linguistic feature increased the reconstruction accuracy compared to the previous model.

$$\text{LMM}_2 : \text{Accuracy} \sim 1 + (1|\text{subject})$$

We compared LMMs with and without Hemisphere effect using R's anova() function. LMM comparison showed that Hemisphere (LMM1 - LMM2 $\Delta \chi^2$ = 4.03, p<0.05, LLM1 Bayesian Information Criterion (BIC): −3331.5, LMM2 BIC: −3329.5) predicted the averaged reconstruction accuracies in Dutch stories but not in French stories. We reported the results of LMM1 in *Table 4* and *Table 5*.

**Table 2.** LMM results of reconstruction accuracies for Dutch stories.

|  | Estimate | Std. Error | Df | t value | Pr(>|t|) | |
|---|---|---|---|---|---|---|
| (Intercept) | 2.86E-03 | 2.38E-04 | 23.1 | 11.99 | 2.14E-11 | *** |
| Phoneme Onset – Acoustic | 5.50E-05 | 1.49E-05 | 92.0 | 3.70 | 3.63E-04 | *** |
| Phoneme Surprisal – Phoneme Onset | 7.90E-05 | 1.49E-05 | 92.0 | 5.32 | 7.41E-07 | *** |
| Phoneme Entropy – Phoneme Surprisal | 1.33E-04 | 1.49E-05 | 92.0 | 8.94 | 3.82E-14 | *** |
| Word Frequency - Phoneme Entropy | 1.48E-04 | 1.49E-05 | 92.0 | 9.96 | 2.84E-16 | *** |

In Dutch stories, linguistic features increased the reconstruction accuracy mostly on the left hemisphere however in French stories only Phoneme Onset and Entropy slightly increased the reconstruction accuracies and we couldn't find any significant lateralization effect.

## Effect of sentence context on neural tracking

To investigate the second question, how do higher level cues (sentence and discourse constraint embodied by word entropy) interact with lower level cues (acoustic- and phoneme-level features), words in each story are grouped into low and high entropy conditions. TRFs including all features were estimated for each condition and language on each hemisphere. We compared the reconstruction accuracies of high and low entropy words by subtracting the reconstruction accuracy of the model which has all features except phoneme features from the full model which includes all features. Similarly, to isolate the effect of contextual constraint on acoustic edges, we compared the averaged reconstruction accuracies by subtracting the reconstruction accuracies of the model which has all features except acoustic edges from the full model which includes all features. To compare the reconstruction accuracies averaged over all brain sources in each hemisphere, we fitted a linear mixed model with random intercept for subjects and a random slope for word entropy, language and hemisphere. The LMM with a random slope for all effects did not converge, so we fitted a random slope for language, hemisphere. As we were interested in the interaction between language and word entropy, we evaluated whether adding language and word entropy interaction increased predictive accuracy. Model comparison between a model with and without interaction was done with ANOVA. The formula of the LMM reads:

$$\text{LMM1}: \text{Accuracy} \sim \text{Language} + \text{Word Entropy} + \text{Hemisphere} + \text{Word Entropy} * \text{Language} + (1 + \text{Language} + \text{Hemisphere} \mid \text{Subject})$$

$$\text{LMM2}: \text{Accuracy} \sim \text{Language} + \text{Word Entropy} + \text{Hemisphere} + (1 + \text{Language} + \text{Hemisphere} \mid \text{Subject})$$

Model comparison showed that language and word entropy interaction predicted the averaged reconstruction accuracies of phoneme features (LMM1 vs LMM2: $\Delta \chi^2$ = 15.315, p<0.00001) and also acoustic edges (LMM1 vs LMM2: $\Delta \chi^2$ = 33.92, p<0.00001). We found a significant main effect both for Language (p<0.001) and Word Entropy (p<0.0001), and interaction between them (p<0.0001). There was a significant main effect of Hemisphere for Acoustic Edges (p=0.045) but not for Phoneme

**Table 3.** LMM results of reconstruction accuracies for French stories.

|  | Estimate | Std. Error | Df | t value | Pr(>|t|) | |
|---|---|---|---|---|---|---|
| (Intercept) | 2.09E-03 | 2.09E-04 | 23.0 | 10.00 | 7.54E-10 | *** |
| Phoneme Onset – Acoustic | 1.86E-05 | 6.34E-06 | 92.0 | 2.94 | 4.15E-03 | ** |
| Phoneme Surprisal – Phoneme Onset | 1.93E-05 | 6.34E-06 | 92.0 | 3.04 | 3.09E-03 | ** |
| Phoneme Entropy – Phoneme Surprisal | 2.88E-05 | 6.34E-06 | 92.0 | 4.54 | 1.70E-05 | *** |
| Word Frequency - Phoneme Entropy | 1.76E-05 | 6.34E-06 | 92.0 | 2.78 | 6.54E-03 | ** |

****<0.0001, ***<0.001, **<0.01, *<0.05.

**Table 4.** LMM results of accuracy improvement by linguistic features for Dutch stories.

|  | Estimate | Std. Error | df | t value | Pr(>|t|) |  |
|---|---|---|---|---|---|---|
| (Intercept) | 4.22E-05 | 6.83E-06 | 31.58 | 6.18 | 6.83E-07 | *** |
| hemisphere_right | −1.06E-05 | 5.26E-06 | 167.00 | −2.02 | 4.55E-02 | * |

****<0.0001, ***<0.001, **<0.01, *<0.05.

Features. Interaction of reconstruction accuracies averaged over whole brain between language and word entropy are shown in *Figure 3A* for acoustic edges and 3-B for phoneme features. LMM results are shown in *Table 6* and *Table 7*.

To examine the contribution of each feature to tracking performance, we compared the TRFs of each feature by running a mass-univariate repeated measures ANOVA on source data to see the main effect of language and word entropy, while also modeling their interaction. Before the statistical analysis, we took the power of TRF weights and smoothed for 2 voxels (Gaussian window, SD = 14 mm) to compensate for head movements of participants. The multiple comparisons problem was handled with a cluster-level permutation test across space and time with 8000 permutations (*Oostenveld et al., 2011*). *Figure 3C, D, E and F* shows the averaged TRFs over all source points of acoustic edge and phoneme features (phoneme onset, phoneme surprisal, and phoneme entropy). For all TRF components, we found a significant main effect of language, word entropy and interaction between them on both hemispheres. Significant source clusters of main effects and interactions for each feature are shown in *Figure 3G*. To show each contrast on the same scale, percentage of power of weights was calculated for each hemisphere.

Power of weights of high entropy words are greater than low entropy word in each speech feature TRF. As opposed to phoneme features, in acoustic edge TRF, power of weights in French stories are greater than in Dutch stories. Interaction between language and word entropy ends earlier for acoustic edges compared to phoneme features. Acoustic Edge TRF peaks around 80ms and 200ms for both languages, whereas phoneme features TRF has a peak around 80ms for both languages but there is a second peak between 200ms and 600ms in Dutch stories.

We validated our analysis on the next 4 Dutch story parts and found the same effects with the reconstruction accuracies and weights of TRFs (*Table 8*, *Table 9* and *Figure 4*).

We also run new models with changing time lags and computed the model accuracy improvement by each feature (See Models with varying time lags in Materials and methods Section) *Figures 5 and 6* shows the reconstruction accuracy improvement by each feature for each time window. On x axis, accuracy of each time windows is shown on their center time. For example, for the window between −100ms and 0ms, it's shown on t=0.05 s. Red bar below shows the time intervals when there was a significant interaction between language and word entropy. Analysis results shows that word entropy modulated the encoding of phoneme onset more in the comprehended language between 200 and 750ms in LH and RH, between 150 and 650ms in LH and between 0 and 600ms in RH for phoneme surprisal, and it was between 550ms and 650ms in LH for phoneme entropy.

## Discussion

In this study, we investigated whether phoneme-level features are tracked over and above acoustic features and acoustic edges, and whether word entropy affects the tracking of acoustic- and phoneme-level features as a function of language comprehension. We used linear encoding models to quantify the degree of neural tracking while native Dutch speakers listened to audiobooks in Dutch

**Table 5.** LMM results of accuracy improvement by linguistic features for French stories.

|  | Estimate | Std. Error | df | t value | Pr(>|t|) |  |
|---|---|---|---|---|---|---|
| (Intercept) | 5.16E-06 | 3.05E-06 | 41.24 | 1.69 | 9.80E-02 | . |
| hemisphere_right | −1.52E-06 | 3.10E-06 | 167.00 | −0.49 | 6.26E-01 |  |

****<0.0001, ***<0.001, **<0.01, *<0.05.

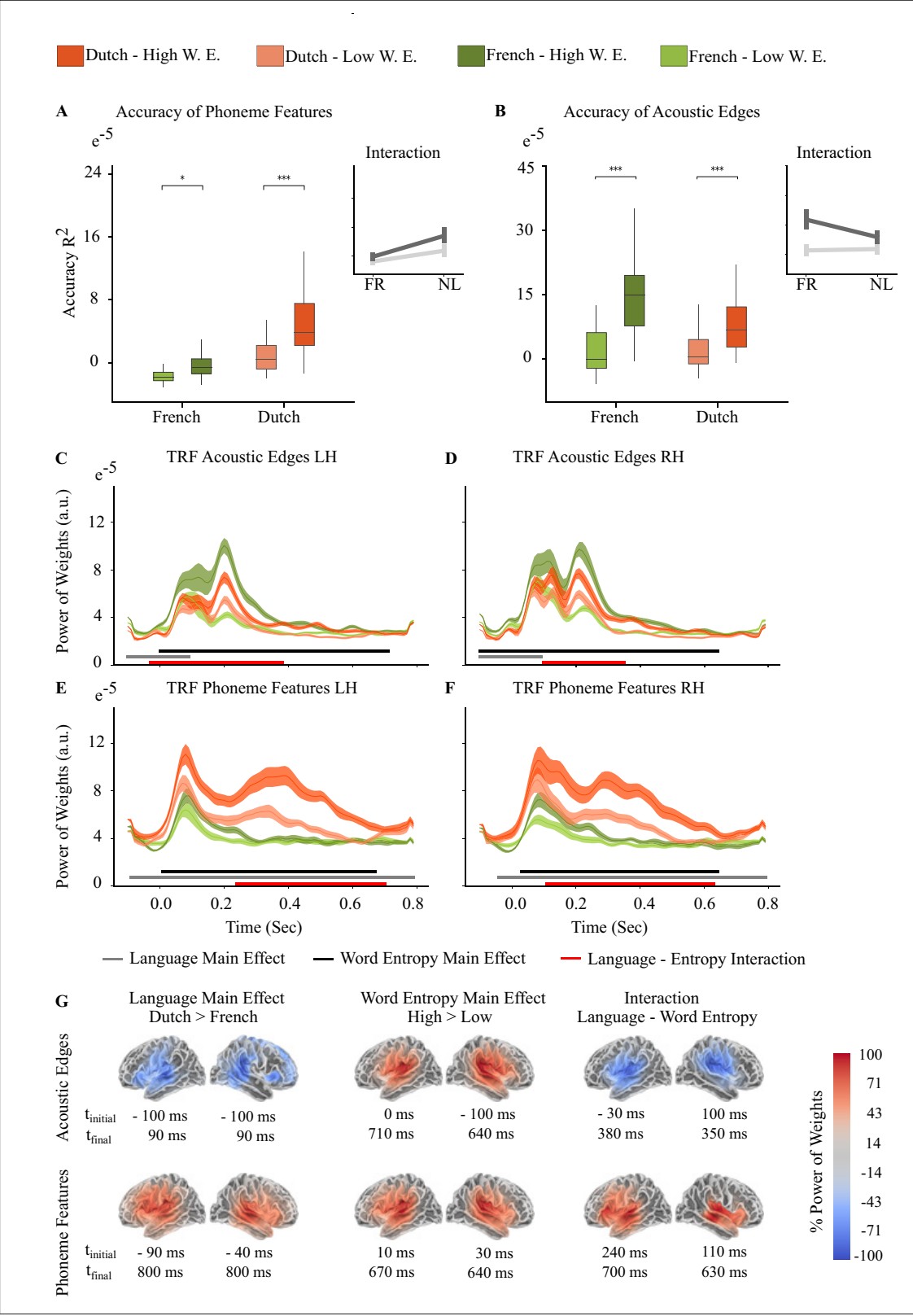

**Figure 3.** First 4 Dutch Story Parts (n=24). Light orange and light green represent Low Word Entropy condition, dark orange and dark green represent High Word Entropy condition for Dutch and French stories, respectively. (**A**) Reconstruction accuracy interaction between word entropy and language for acoustic features (**B**) Reconstruction accuracy interaction between word entropy and language for phoneme features (Braces in Figure A and B indicate the significant different between high and low entropy word conditions, ****<0.0001, ***<0.001, **<0.01, *<0.05. Error bars shows within subject

*Figure 3 continued on next page*

*Figure 3 continued*

standard errors.) (**C-D**) Acoustic Edge TRFs on left hemisphere (LH) and right hemisphere (RH) (**E-F**) Phoneme Features TRFs on LH and RH. Lines on the graphs in Figure C-F show the mean and shaded areas show the standard error of the mean. (**G**) Sources where the main effect of Language and Word Entropy, and interaction are found.

The online version of this article includes the following figure supplement(s) for figure 3:

**Figure supplement 1.** First 4 Dutch Stories (n=24).

and French. Participants were familiar with the prosodic and phonotactic statistical regularities of the French language, but did not comprehend the stories, allowing for the contrast of speech processing during and in the absence of comprehension in a statistically familiar auditory and perceptual context. First, we examined whether phoneme features are tracked by the brain signal even when acoustic features are controlled. Results showed that all phoneme features (i.e. phoneme onset, phoneme surprisal, phoneme entropy) and word frequency increased the averaged reconstruction accuracy for both Dutch and French stories when acoustic features were controlled, extending the findings of previous studies (*Brodbeck et al., 2018*; *Gillis et al., 2021*; *Brodbeck et al., 2022*). An interaction between language and encoding model showed that reconstruction accuracy improvement from the addition of phoneme-level features and word frequency was greater in Dutch stories, suggesting that when language is comprehended, internally-generated linguistic units, e.g., phonemes, are tracked over and above acoustic edges (*Meyer et al., 2020*). Our findings contradict the results of *Daube et al., 2019*, who used articulatory features and phoneme onsets as phonetic features and showed that phoneme-locked responses can be explained by the encoding of acoustic features. It is possible that information theoretic metrics, such as surprisal and entropy, are better suited to capture the transformation of acoustic features into abstract linguistic units.

Comparison of the reconstruction accuracies on the source level showed that, for Dutch stories, accuracy improvement by the linguistic features were left lateralized. However, for uncomprehended French stories, we couldn't find a significant lateralization effect. The Dutch-speaking participants we selected were familiar with the acoustics and phonotactics of French, and thus with the statistical distribution of those features, even though they cannot comprehend the language (as evidenced by their chance-level behavioral performance, educational background, and self-report). This comparison of neural signals in acoustically-familiar contexts set the bar higher for detecting neural sensitivity to phoneme-level features in a comprehended language, over and above any general statistical sensitivity the brain might show during perception and perceptual adaptation. The slight increase in reconstruction accuracy on the phoneme-level in French is likely due to the participants' general familiarity with the phonotactic properties of the language, though, given the duration of the experiment, participants also could have learned the phonotactic distribution of the stories, as well as become sensitive to high-frequency words that they may recognize, but which do not lead to sentence and discourse comprehension (e.g. the most frequent words in the stories were *le, la, un, une, les, des* – all forms of the definite and indefinite determiner *the/a*); statistical regularities can be acquired very quickly even by naive listeners (*Saffran et al., 1996*).

We performed a word-level analysis to examine the second question: do higher level linguistic constraints, as formed during comprehension, interact with the neural encoding lower-level information (acoustic and phonemic features) under normal listening conditions? We operationalized constraint as word entropy, or the uncertainty around word predictability given the previous 30 words. When the word entropy is high, the next word is less reliably predictable given the previous

**Table 6.** LMM results of reconstruction accuracies for Phoneme features.

|  | Estimate | Std. Error | df | t value | Pr(>\|t\|) | |
|---|---|---|---|---|---|---|
| (Intercept) | 8.68E-04 | 8.17E-05 | 26.4 | 10.62 | 5.11E-11 | *** |
| French | −8.49E-04 | 6.75E-05 | 35.1 | −12.58 | 1.46E-14 | *** |
| Low Word Entropy | −2.69E-04 | 4.21E-05 | 118.0 | −6.38 | 3.65E-09 | *** |
| Right Hemisphere | 1.22E-04 | 9.96E-05 | 23.0 | 1.23 | 2.33E-01 | |
| French: Low Word Entropy | 3.74E-04 | 5.96E-05 | 118.0 | 6.29 | 5.69E-09 | *** |

**Table 7.** LMM results of reconstruction accuracies for Acoustic Edges.

|  | Estimate | Std. Error | df | t value | Pr(>|t|) |  |
| --- | --- | --- | --- | --- | --- | --- |
| (Intercept) | 6.83E-05 | 1.66E-05 | 32.8 | 4.11 | 2.45E-04 | *** |
| French | 8.85E-05 | 1.10E-05 | 141.0 | 8.03 | 3.48E-13 | *** |
| Low Word Entropy | −6.08E-05 | 1.10E-05 | 141.0 | −5.52 | 1.61E-07 | *** |
| Right Hemisphere | 4.11E-05 | 1.94E-05 | 23.0 | 2.12 | 4.49E-02 | * |
| French: Low Word Entropy | −9.54E-05 | 1.56E-05 | 141.0 | −6.12 | 8.76E-09 | *** |

****<0.0001, ***<0.001, **<0.01, *<0.05.

words, so the context is less constraining. However, word frequency and predictability are known to be highly correlated (*Cohen Priva and Jaeger, 2018*; in our study, the correlation between entropy and word frequency was French stories: $R$=0.336, p<0.00001; Dutch stories: $R$=0.343, p<0.00001). To control for the responses correlated with word frequency, we also added word frequencies in our full model and compared the reconstruction accuracy improvement explained by acoustic and phoneme features. When we compared the reconstruction accuracies for low and high entropy words, we found that tracking performance of phoneme and acoustic features in high entropy words was higher in Dutch stories. This suggests that sentence and discourse constraint, as expressed in word entropy, modulates the encoding of sub-lexical representations, consistent with top-down models of language comprehension. Results showed that when context was more reliable (i.e. on low entropy words) the neural contribution of acoustic and phoneme-level features is downregulated, inhibited, or not enhanced. When sentence- and discourse-level context constrains lexical representations, sub-lexical representations are inhibited by gain modulation as they are no longer as important for thresholding (*Martin, 2016*; *Martin, 2020*).

We also found an interaction between language comprehension and word entropy in the opposite direction for acoustic edges and phoneme features: the effect of sentence and discourse-level context was larger on phoneme features for a comprehended language, and it was larger on acoustic features for the uncomprehended language. In the uncomprehended language, low entropy words were also the most frequent words presented, such as *le, la, un, une*. Modulation of acoustic edges by context in this situation could be related to statistical 'chunking' of the acoustic signal for frequent words, essentially reflecting recognition of those single function words in the absence of language comprehension. In contrast, when a language is understood, contextual information strongly modulates the tracking of phoneme features. Yet, when the next word is predicted from the context, phoneme features are not as informative as when the context is less containing, and they are downregulated or suppressed such that their encoding does not contribute as much to the composition of the neural signal.

To investigate how language comprehension and word entropy modulated acoustic and phoneme-level linguistic features as they unfolded in time, we analyzed the weights of the TRFs. The amplitude of the power of weights were greater for phoneme-level features in Dutch stories than in French stories, however it was lower for acoustic edges. This suggests that when the language is not comprehended, acoustic features may be more dominant in the neural response, as linguistic features simply are not available. This pattern of results suggest that language comprehension might suppress neural tracking of acoustic edges, or it suppresses the neural representation of edges. If this is so, then language comprehension can be seen as a neural filter over sensory and perceptual input in service of

**Table 8.** LMM results of reconstruction accuracies for Phoneme Features (Next 4 Dutch Story Parts).

|  | Estimate | Std. Error | Df | t value | Pr(>|t|) |  |
| --- | --- | --- | --- | --- | --- | --- |
| (Intercept) | 8.14E-05 | 7.03E-06 | 39.1 | 11.57 | 3.36E-14 | *** |
| French | −8.00E-05 | 5.62E-06 | 141.0 | −14.23 | 2.00E-16 | *** |
| Low Word Entropy | −5.00E-05 | 5.62E-06 | 141.0 | −8.89 | 2.61E-15 | *** |
| Right Hemisphere | −5.32E-06 | 7.61E-06 | 23.0 | −0.70 | 4.91E-01 |  |
| French: Low Word Entropy | 3.67E-05 | 7.95E-06 | 141.0 | 4.61 | 8.87E-06 | *** |

**Table 9.** LMM results of reconstruction accuracies for Acoustic Edges (Next 4 Dutch Story Parts).

|  | Estimate | Std. Error | df | t value | Pr(>|t|) |  |
|---|---|---|---|---|---|---|
| (Intercept) | 8.38E-05 | 1.70E-05 | 32.0 | 4.93 | 2.44E-05 | *** |
| French | 7.70E-05 | 1.09E-05 | 141.0 | 7.06 | 7.15E-11 | *** |
| Low Word Entropy | −8.18E-05 | 1.09E-05 | 141.0 | −7.49 | 6.76E-12 | *** |
| Right Hemisphere | 3.30E-05 | 1.92E-05 | 23.0 | 1.72 | 9.89E-02 |  |
| French: Low Word Entropy | −7.44E-05 | 1.54E-05 | 141.0 | −4.82 | 3.66E-06 | *** |

****<0.0001, ***<0.001, **<0.01, *<0.05.

the transformation of that input into linguistic structures. A similar effect was also shown for increasing speech rate; tracking performance of linguistic features decreased and tracking performance of acoustic features increased with decreasing intelligibility (*Verschueren et al., 2022*).

We also found that phoneme features generated a peak at around 80ms and between 200 and 600ms in the comprehended language, whereas in an uncomprehended language, they only peaked at around 80ms. This dual-peak pattern could be attributable to the internal generation of morphemic units or to lexical access in the comprehended language. Similar to the results we found in reconstruction accuracies, comparison of low and high entropy word TRFs showed that when the word entropy was higher (viz., in a low-constraining sentence and discourse context), the weights of acoustic edges and phoneme-level features were higher. This result suggests that when the higher level cues are not available or reliable, lower-level cues at the phoneme- and acoustic-level may be upregulated or enhanced, or allowed to propagate in order to comprise more of the neural bandwidth. While our results are consistent with *Molinaro et al., 2021*. – we provide support for a cost minimization perspective rather than the perception facilitation perspective discussed in Molinaro et al. – it is important to note that Molinaro et al. only examined the tracking of acoustic features, specifically the speech envelope, using the Phase Locking Value, and did not examine the contribution of lower-level linguistic features. Secondly, Molinaro et al. use a condition-based experimental design in contrast to our naturalistic stimulus approach. In our study, our aim was to investigate the dynamics of encoding both acoustic and linguistic features, and we utilized a multivariate linear regression method on low and high constraining words which 'naturally' occurred in our audiobook stimulus across languages. Our results revealed a trade-off between the encoding of acoustic and linguistic features that was dependent on the level of comprehension. Specifically, in the comprehended language, the predictability of the following word had a greater influence on the tracking of phoneme features as opposed to acoustic features, while in the uncomprehended language, this trend was reversed. Similarly, *Donhauser and Baillet, 2020* showed that when sentence context is not reliable, phonemic features (i.e. phoneme surprisal and entropy) are enhanced. Here we examined the effects word entropy on acoustic and phonemic representations in comprehended and uncomprehended languages; we found that comprehension modulates acoustic edges for a shorter period of time than phoneme features when language is comprehended. The interaction between language comprehension and word entropy for acoustic edge encoding was located around auditory cortex in both hemispheres, whereas for phoneme features, it was located in left frontal cortex and right temporal cortex. The presence of a later interaction for phoneme onsets compared to acoustic edges could be due to the sustained representations of phonemes until lexical ambiguity is resolved (*Gwilliams et al., 2020*) – when the sentence context is not very informative, phonemic representations transformed from acoustic features may persist or play a larger role in the neural response until the word is recognized or until a structure is built.

In this study, to disassociate the effects of word entropy and word frequency, we modeled word frequency with an assumption that entropy and frequency have linearly additive effects; however, previous studies have shown that they interact during late stages of word processing (*Fruchter and Marantz, 2015*; *Huizeling et al., 2022*). In a naturalistic listening paradigm, even though it is an instrumental way to study language comprehension, it is not fully possible to control for word frequency or to completely dissociate it from any sentence context effects.

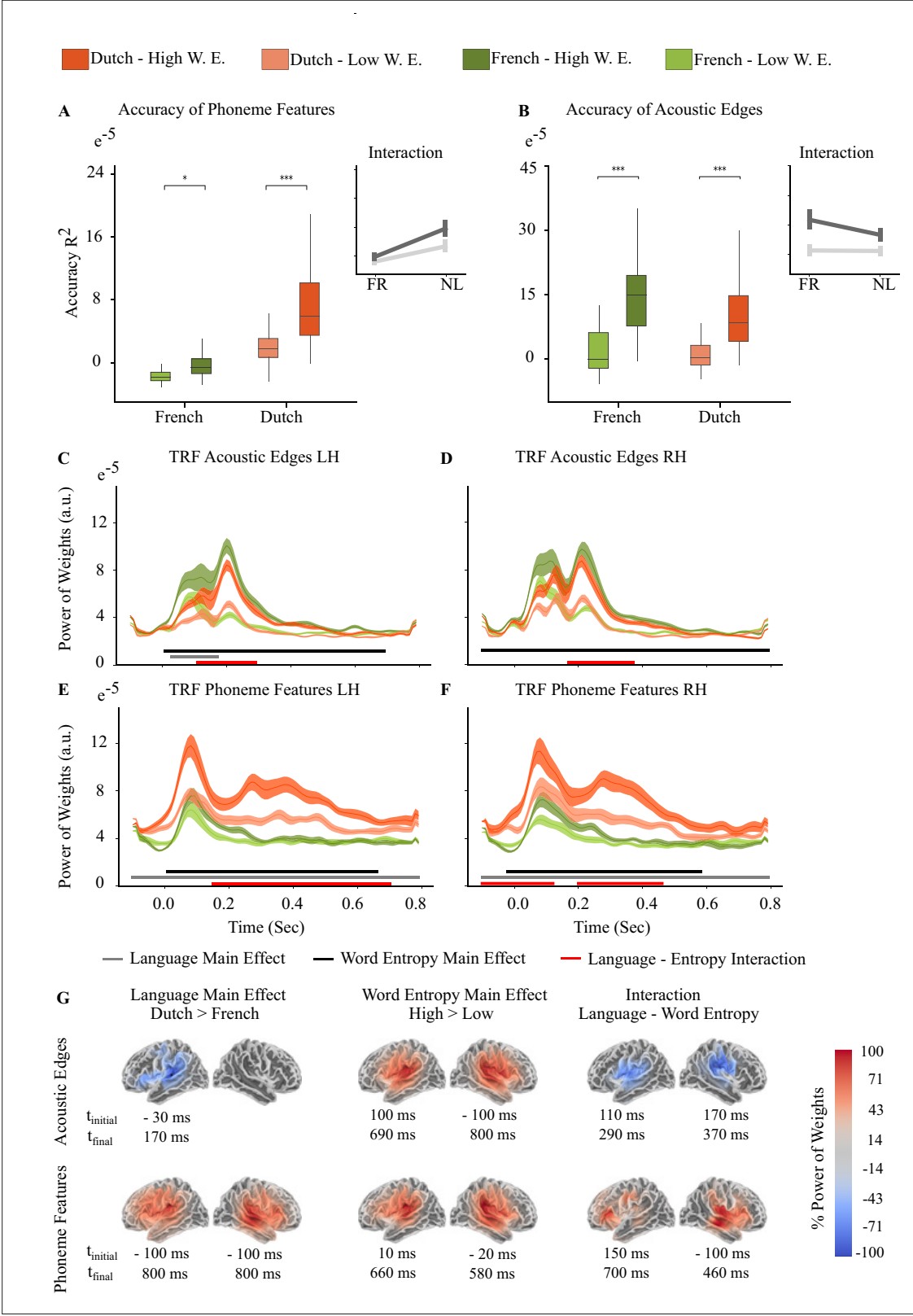

**Figure 4.** Second 4 Dutch Story Parts (n=24). Light orange and light green represent Low Word Entropy condition, dark orange and dark green represent High Word Entropy condition for Dutch and French stories, respectively. (**A**) Reconstruction accuracy interaction between word entropy and language for acoustic features. (**B**) Reconstruction accuracy interaction between word entropy and language for phoneme features. (Braces in Figure A and B indicate the significant different between high and low entropy word conditions, ****<0.0001, ***<0.001, **<0.01, *<0.05. Error bars shows

*Figure 4 continued on next page*

*Figure 4 continued*

within subject standard errors.) (**C-D**) Acoustic Edge TRFs on left hemisphere (LH) and right hemisphere (RH). (**E-F**) Phoneme Features TRFs on LH and RH. Lines on the graphs in Figure C-F show the mean and shaded areas show the standard error of the mean. (**G**) Sources where the main effect of Language and Word Entropy, and interaction are found.

The online version of this article includes the following figure supplement(s) for figure 4:

**Figure supplement 1.** Next 4 Dutch Stories (n=24).

Another limitation of our study is that linear regression modeling does not allow the dissociation of the weights of highly correlated features. When we compared the TRF weights of phoneme onset, surprisal and entropy separately between the set of the first four and the second four Dutch story parts, we saw that TRF weights of these features showed differences (*Figure 3—figure supplement 1*, *Figure 4—figure supplement 1*). However, acoustic-edge features and averaged phoneme-level features were highly consistent between different story parts, as acoustic and averaged phoneme features are not as correlated as phoneme onset, phoneme surprisal and phoneme entropy (*Figure 3* and *Figure 4*). Thus, it would be problematic to interpret how language comprehension and sentence and discourse level constraint affects these features separately. As alternative solution to this problem, we also fitted different models with varying time lags separated by 50ms with a 100ms sliding window between –100 and 800ms and also compared the reconstruction accuracies of these models instead of TRF weights. Results showed that word entropy modulated the encoding of phoneme onset and phoneme surprisal for a longer time, starting earlier compared to phoneme entropy.

## Conclusion

In this study, we show that modeling phoneme-level linguistic features in addition to acoustic features better reconstructs the neural tracking response to spoken language, and that this improvement is even more pronounced in the comprehended language, likely reflecting the transformation of acoustic features into phoneme-level representations when a language is fully comprehended. Although acoustic edges are important for speech tracking, internally generated phonemes are more strongly encoded in comprehended language. This suggests that language comprehension can be seen as a neural filter over acoustic edges of the speech signal in the service of transforming sensory input into abstract linguistic units. We demonstrated that low sentence- and discourse-level constraints enhanced the neural encoding of both acoustic and phoneme features. When language is not comprehended, the neural encoding of acoustic features was stronger, likely due to the absence of comprehensive lexical access and interpretation, including higher level linguistic structure formation, in addition to the recognition of single, highly frequent function words (viz., the/a) in isolation. When a language is comprehended from speech, phoneme features were more strongly encoded in the neural response compared to when it is not comprehended. Only in the comprehended language, phoneme features aligned with the phase of neural signal between 200 and 600ms, and this is effect was stronger when the context was less constraining and more information must be extracted from the sensory signal. This pattern of results may reflect the persistence of phoneme-related neural activity before a word is recognized and phoneme-level information can be inhibited. Relying more on low-level features when high-level cues are not reliable, and suppressing low-level information when contextual cues are informative could be a strategy for the brain to utilize its resources in an efficient way toward its behavioral goal. In summary, our results support an account of language comprehension where the flexible modulation of the acoustic and phonemic features by lexical, sentential, and discourse-level information is instrumental in the transformation of sensory input into interpreted linguistic structure and meaning.

## Materials and methods
### Participants

We collected MEG data from 24 participants between 18 and 58 years old (average age: 31.17 years, 18 F and 6 M) while they were listening audiobooks in Dutch and French. They were all right handed native Dutch speakers with either no or very little French proficiency. Four of the participants reported that they can only comprehend a full French sentence only if it is spoken very slowly or it is a very

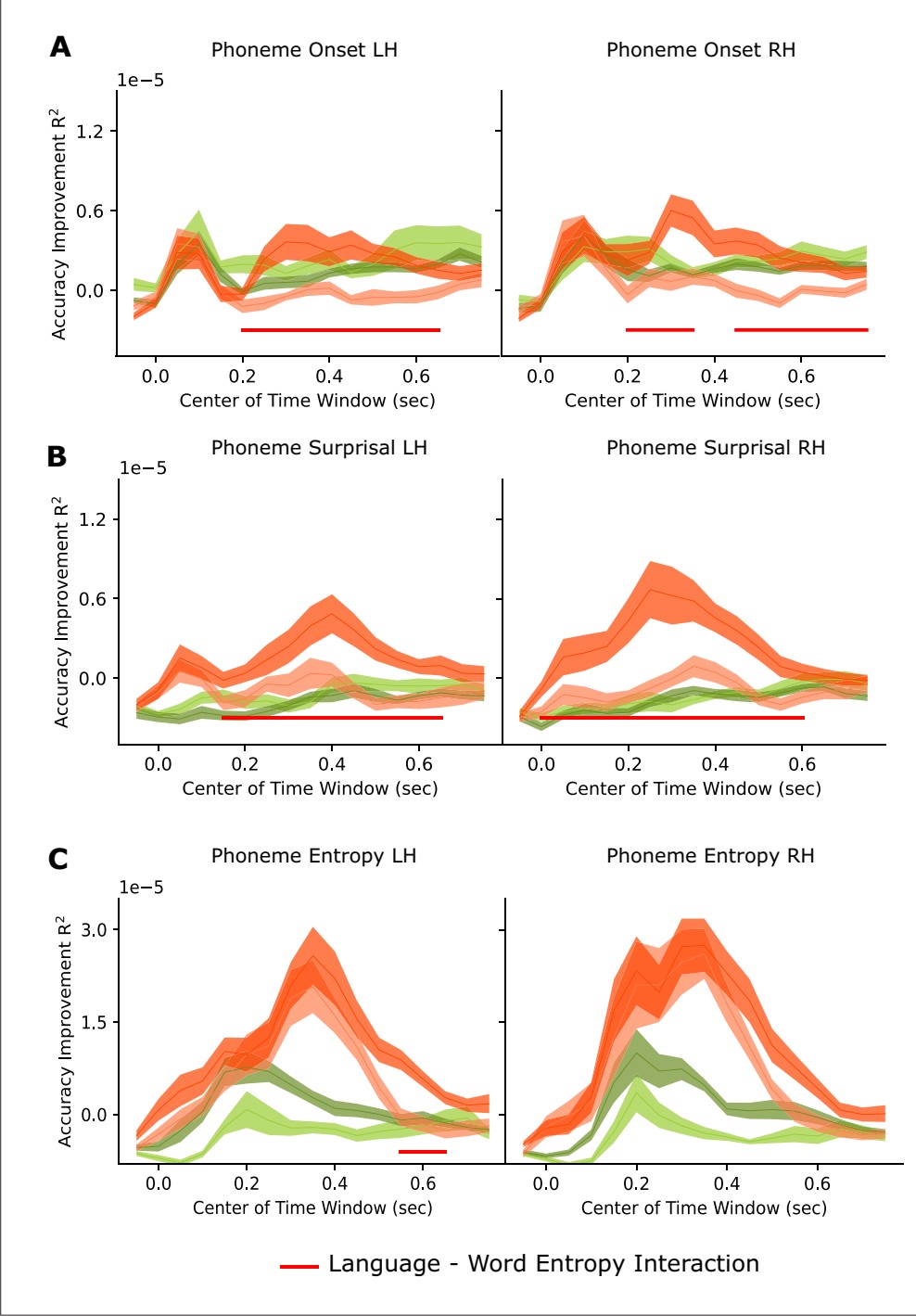

**Figure 5.** Accuracy improvement by each linguistic feature calculated by subtracting the model accuracy of previous model from the model which also has the feature of interest for each time window in High and Low Word Entropy Conditions (n=24). (**A**) Phoneme Onset (Left Hemisphere – LH on the left, Right Hemisphere RH on the right). (**B**) Phoneme Surprisal. (**C**) Phoneme Entropy. Lines on the graphs in Figure A-C show the mean and shaded areas show the standard error of the mean.

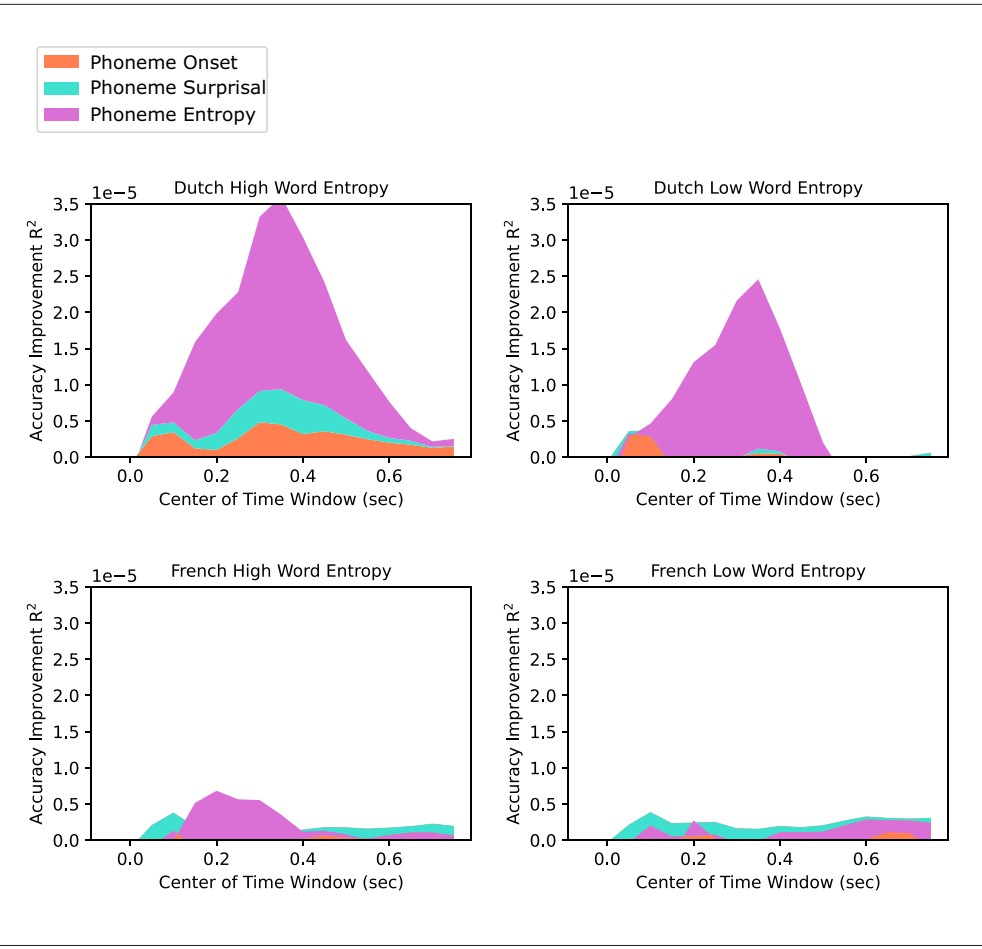

**Figure 6.** Accuracy improvement by each linguistic feature calculated by subtracting the model accuracy of previous model from the model which also has the feature of interest for each time window (n=24). High (Left) and Low (Right) Word Entropy Conditions for French (Below) and Dutch (Top) Stories.

simple sentence. Rest of the participants reported they cannot comprehend a full sentence. This study was approved by the ethical Commission for human research Arnhem/Nijmegen (project number CMO2014/288). Participants were reimbursed for their participation.

## Stimuli

The stimuli (*Table 10*) consisted of one story by Hans Christian Andersen, two stories by the Brothers Grimm in Dutch and one story by Grimm, one by E.A. Poe, and one by Andersen in French. (*Kearns, 2015*; *Hart, 1971*) All stories are divided into 5–6 min story parts (9 story parts in Dutch, 4 story parts in French). The stories are presented in a randomized order to the participants. After each part, participants were asked to answer five multiple-choice comprehension questions. For our analysis, we used the first four story parts in each language to balance the length of data in each language. Then we repeated the same analysis for second four part of Dutch stories.

## Data acquisition

Brain activity of participants were recorded using magnetoencephalography (MEG) with a 275-sensor axial gradiometer system (CTF Systems Inc) in a magnetically shielded room. Before presenting each story part, 10 s of resting state data were recorded. All stimuli were presented audibly by using the Psychophysics Toolbox extensions of Matlab (*Brainard, 1997*; *Pelli, 1997*; *Kleiner et al., 2007*) while participants were fixating a cross in the middle of the presentation screen. MEG data were acquired at a sampling frequency of 1200 Hz. Head localization was monitored during the experiment using marker coils placed at the cardinal points of the head (nasion, left and right ear canal) and head

**Table 10.** Stimuli.

| Story Part | Language | Duration | Speaker | Parts used in analysis |
|---|---|---|---|---|
| Anderson_S01_P01 | NL | 4 min 58 s | Woman 1 | Dutch Part 1 |
| Anderson_S01_P02 | NL | 5 min 17 s | Woman 1 | Dutch Part 1 |
| Anderson_S01_P03 | NL | 4 min 49 s | Woman 1 | Dutch Part 1 |
| Anderson_S01_P04 | NL | 5 min 50 s | Woman 1 | Dutch Part 1 |
| Grimm_23_1 | NL | 5 min 3 s | Woman 2 | Dutch Part 2 |
| Grimm_23_2 | NL | 5 min 32 s | Woman 2 | Dutch Part 2 |
| Grimm_23_3 | NL | 5 min 2 s | Woman 2 | Dutch Part 2 |
| Grimm_20_1 | NL | 6 min 6 s | Woman 2 | Dutch Part 2 |
| ANGE_part1 | FR | 4 min 34 s | Woman 3 | French Part 1 |
| BALL_part1 | FR | 4 min 58 s | Woman 3 | French Part 1 |
| EAUV_part1 | FR | 5 min 43 s | Man 1 | French Part 1 |
| EAUV_part2 | FR | 6 min 1 s | Man 1 | French Part 1 |

position was corrected before each story part presentation to keep it at the same position as at the beginning of the experiment. Bipolar Ag/AgCl electrode pairs were used to record electrooculogram (EOG) and electrocardiogram (ECG). In addition to MEG data, we also acquired T1-weighted structural MR images using a 3T MAGNETOM Skyra scanner (Siemens Healthcare, Erlangen, Germany). Lastly, three-dimensional coordinates of each participants head surface was measured using a digitizing pen system (Polhemus Isotrak system, Kaiser Aerospace Inc).

## MEG data preprocessing

MEG data were analyzed using mne-python (version 0.23.1). First, data were annotated to exclude the response parts from the rest of the analysis, then filtered between 0.5 and 40 Hz with one-pass, zero-phase, non-causal FIR filter using the default settings of mne-python. Bad channels were removed using mne implementation of Maxwell filtering, and removed channels were interpolated. Then data were resampled to 600 Hz and ocular and cardiac artifacts were removed with independent components analysis. Each story part segments were cropped from preprocessed data and source localization was done separately for each part. Before the source localization, each part was low pass filtered at 8 Hz with one-pass, zero-phase, non-causal FIR filter.

Individual head models were created for each participant with their structural MR images with Freesurfer (surfer.nmr.mgh.harvard.edu) and were co-registered to the MEG coordinate system with mne coregistration utility. A surface-based source space was computed for each participant using fourfold icosahedral subdivision. Cortical sources of the MEG signals were estimated using noise-normalized minimum norm estimate method, called dynamic statistical parametric map (dSPM). Orientations of the dipoles were constrained to be perpendicular to the cortical surface. Resting state data before the presentation of each story part (130 s in total) were used to calculate the noise covariance matrix. Lastly, source time courses were resampled at 100 Hz.

## Predictor variables
### Acoustic features
Acoustic features (8 band gammatone spectrogram and an 8-band acoustic onset spectrogram, both covering frequencies from 20 to 5000 Hz in equivalent rectangular bandwidth (ERB) space) for each story part were generated using Eelbrain toolbox (*Brodbeck et al., 2021*).

### Phoneme onsets
Phoneme onsets were extracted from the audio files of the stories automatically using the forced alignment tool from WebMAUS Basic module of the BAS Web Services (*Schiel, 1999*; *Strunk et al., 2014*).

## Phoneme surprisal and entropy

Probabilities of each phoneme in a given word was calculated according to the probability distribution over the lexicon of each language weighted by the occurrence frequency of each word. When each phoneme unfolds in a word, it reduces the number of possible words in the cohort and generates a subset of cohort$_i$. Conditional probability of each phoneme Ph$_i$ given the previous phoneme equals to the ratio of total frequencies of the words in the remaining cohort to previous cohort.

$$P\left(Ph_i|Ph_{(i-1)}\right) = \sum_{word \,\in\, cohort_i} freq_{word}\left(i\right) / \sum_{word \,\in\, cohort_{(i-1)}} freq_{word}\left(i-1\right)$$

The surprisal of phoneme Ph$_i$ is inversely related to the likelihood of that phoneme.

$$S\left(Ph_i|Cohort_{(i)}\right) = -log_2\left(P\left(Ph_i\right)\right)$$

The entropy of phoneme Ph$_i$ quantifies the uncertainty about the next phoneme Ph$_{i+1}$. It is calculated by taking the average of expected surprisal values of all possible phonemes.

$$E\left(Ph_i|Cohort_{(i)}\right) = \sum_{Ph}^{All\ phonemes} -P\left(Ph|cohort_{(i-1)}\right) * log_2\left(P\left(Ph|cohort_{(i-1)}\right)\right)$$

To calculate the probabilities of phonemes, we used SUBTLEX-NL dictionary (**Keuleers et al., 2010**) for Dutch stories and Lexique383 dictionary (9_freqfilms from subtitles) for French stories. Both dictionaries were filtered to eliminate the words which contains nonalphabetic characters. French dictionary had 92.109 words, so the same number of most frequent words also selected from Dutch dictionary. Words in the dictionaries are transformed into phonemic transcription by using the g2p module of The BAS Web Services (**Schiel, 1999**; **Schiel, 2015**).

## Word frequency

Word frequency features were calculated by taking negative logarithm of word frequencies varying between 0 and 1 which were calculated as word occurrence per 1,000,000 words in SUBTLEX-NL dictionary (**Keuleers et al., 2010**) for Dutch stories and Lexique383 dictionary (9_freqfilms from subtitles) for French stories (**New et al., 2001**).

$$Word\ Freq = -log_2\left(Freq_i\right)$$

## Word entropy

To quantify the uncertainty about the next word given previous 30 words, we used the transformer-based language model GPT-2 which were fine tuned for Dutch (**de Vries and Nissim, 2020**) and French (**Louis, 2020**). Entropy values for each word were calculated from the probability distribution generated by the GPT-2 language model.

**Table 11.** Model names and speech features in models.

|  | Spectrogram | Acoustic Edge | Phoneme Onset | Phoneme Surprisal | Phoneme Entropy | Word Frequency |
|---|---|---|---|---|---|---|
| Acoustic | ✓ | ✓ |  |  |  |  |
| Phoneme Onset | ✓ | ✓ | ✓ |  |  |  |
| Phoneme Surprisal | ✓ | ✓ | ✓ | ✓ |  |  |
| Phoneme Entropy | ✓ | ✓ | ✓ | ✓ | ✓ |  |
| Word Frequency | ✓ | ✓ | ✓ | ✓ | ✓ | ✓ |

$$E\left(Word_i|Context_{(i-30:\ i-1)}\right)$$

$$= \sum_{T}^{All\ tokens} -P\left(T|Context_{(i-30:\ i-1)}\right) * log_2\left(P\left(T|Context_{(i-30:\ i-1)}\right)\right)$$

Words in each story part were divided into two conditions; high and low word entropy, so that each condition has equal length of signal in each story part (*Figure 1*).

## Linear encoding models

Four different models were built by incrementally adding each phonemic feature on top of acoustic control features. Model names and features included in models are shown in *Table 11*.

Temporal response functions (TRF) were computed for each model, subject and source using the Eelbrain toolbox (*Brodbeck et al., 2021*). For each model, corresponding speech features were shifted by T lags between –100ms and 800ms from the onset of each phoneme. With 50ms wide Hamming windows at 100 Hz sampling rate that yields T=90 time points. MEG response at time t $\left\{y_i\left(t_n\right)\right\}_{j=1}^{N}$ (N=5,124 virtual current source, i: subject number, $t_n$: time point) was predicted by convolving the TRF with predictor features shifted by T time delays $\left\{x_f\left(t_n - \tau_k\right)\right\}_{f=1}^{F}$ (F: number of speech features in the model). $\beta_{ijf}\left(\tau_k\right)$ is the TRF of $i^{th}$ subject, $j^{th}$ source point, $f^{th}$ speech features at $k^{th}$ latency.

$$y_i^j\left(t_n\right) = \sum_{f=1}^{F}\sum_{k=1}^{T}\beta_{ijf}\left(\tau_k\right)x_f\left(t_n - \tau_k\right)$$

All predictors and MEG signals were normalized by dividing by the absolute mean value. To estimate TRFs, boosting algorithm of Eelbrain toolbox was used to minimize the l1 *error* using a fivefold cross-validation procedure. We used the early stopping from the toolbox. It uses a validation set which is distinct from the test set to stop training when the error starts to increase to prevent overfitting (*Brodbeck et al., 2021*). As there were four French story parts, we only used the first four Dutch story parts. Total duration of French stories was 21 min 17 s and it was 20 min 54 s for Dutch stories. Then we repeated the same analysis with the next four Dutch story parts. Total duration of the next four Dutch stories was 21 min 43 s.

## Model accuracy comparison

To evaluate the effect of adding each feature on top of acoustic features on the reconstruction accuracy, proportion of the explained variance values on each source point were smoothed (Gaussian window, SD = 14 mm) and a linear mixed model with random slope for subjects were fitted for the average reconstruction accuracy values of each model on each hemisphere using the lmer function in the lme4 package for R.

To identify the brain region where a specific feature increased model accuracy, model accuracies on each source point for each model were compared with the model accuracies of previous model that has all other speech features except the features of interest using a mass-univariate two-tailed related sample t-test with threshold-free cluster enhancement (TFCE) (*Smith and Nichols, 2009*).

## Models with varying time lags

We also run new models with changing time lags and computed the model accuracy improvement by each feature by subtracting the reconstruction accuracy of a model from the previous model which does not have the feature of interest as it was done in the first analysis where we showed the additional contribution of each feature to the reconstruction accuracy. We use 17 different time lags separated by 50ms with a 100ms sliding window between –100 and 800ms. We then compared the averaged reconstruction accuracies over whole brain of high and low word entropy conditions of each language (French and Dutch) with a cluster-level permutation test across time with 8000 permutations.

## Acknowledgements

We thank Sanne ten Oever for constructive feedback on the study design, and Ryan MC Law, Ioanna Zioga, Cas Coopmans, and Sophie Slaats for contributing to data acquisition.

# Additional information

### Competing interests

Andrea E Martin: Reviewing editor, *eLife*. The other authors declare that no competing interests exist.

### Funding

| Funder | Grant reference number | Author |
|--------|------------------------|--------|
| Max-Planck-Gesellschaft | Lise Meitner Research Group "Language and Computation in Neural Systems" | Andrea E Martin |
| Max-Planck-Gesellschaft | Independent Research Group "Language and Computation in Neural Systems" | Andrea E Martin |
| Nederlandse Organisatie voor Wetenschappelijk Onderzoek | 016.Vidi.188.029 | Andrea E Martin |

The funders had no role in study design, data collection and interpretation, or the decision to submit the work for publication. Open access funding provided by Max Planck Society.

### Author contributions

Filiz Tezcan, Conceptualization, Data curation, Software, Formal analysis, Investigation, Visualization, Methodology, Writing - original draft, Writing - review and editing; Hugo Weissbart, Conceptualization, Data curation, Supervision, Validation, Visualization, Methodology, Writing - review and editing; Andrea E Martin, Conceptualization, Resources, Formal analysis, Supervision, Funding acquisition, Validation, Methodology, Project administration, Writing - review and editing

### Author ORCIDs

Filiz Tezcan http://orcid.org/0000-0003-3327-0181
Hugo Weissbart http://orcid.org/0000-0003-2820-3865
Andrea E Martin http://orcid.org/0000-0002-3395-7234

### Ethics

Human subjects: Participants performed a screening for their eligibility in the MEG and MRI and gave written informed consent. The study was approved by the Ethical Commission for human research Arnhem/Nijmegen (project number CMO2014/288). Participants were reimbursed for their participation.

### Decision letter and Author response

Decision letter https://doi.org/10.7554/eLife.82386.sa1
Author response https://doi.org/10.7554/eLife.82386.sa2

# Additional files

### Supplementary files

• MDAR checklist

### Data availability

The raw data used in this study are available from the Donders Institute Data Repository (https://doi.org/10.34973/a65x-p009) By the time of the submission of the paper, the following subjects had both MRI and MEG data: sub-001, sub-003, sub-004, sub-008, sub-009, sub-010, sub-011, sub-013, sub-014, sub-015, sub-017, sub-018, sub-019, sub-020, sub-021, sub-023, sub-025, sub-026, sub-027, sub-028, sub-029, sub-030, sub-032, and sub-033. They were included in the analysis. Analysis code is shared on the following link. https://github.com/tezcanf/Scripts_for_publication.git (copy archived at swh:1:rev:23b3c7ad9dcabae84dca2a017ee8260081a9d18c).

The following dataset was generated:

| Author(s) | Year | Dataset title | Dataset URL | Database and Identifier |
|---|---|---|---|---|
| Martin AE | 2023 | Constructing sentence-level meaning: an MEG study of naturalistic language comprehension | https://doi.org/10.34973/a65x-p009 | Donders Institute Data Repository, 10.34973/a65x-p009 |

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
