## [Editor Report]

This study addresses a fundamental aspect of human speech processing: namely, how acoustic and linguistic features interact during comprehension. The authors present convincing evidence that helps elucidate the role of language experience on neural processing, re-weighting processing of speech based on whether a listener understands the language being spoken.

---

## [Decision Letter]

**Decision letter after peer review:**

Thank you for submitting your article "A tradeoff between acoustic and linguistic feature encoding in spoken language comprehension" for consideration by *eLife*. Your article has been reviewed by 3 peer reviewers, one of whom is a member of our Board of Reviewing Editors, and the evaluation has been overseen by Andrew King as the Senior Editor. The reviewers have opted to remain anonymous.

Essential revisions:

1) The aspects of the current results that are consistent with vs. different to prior arguments should be better clarified. As it stands, the theoretical advance of the current paper relative to much current thinking in the field is not always clear.

2) The treatment of the high-entropy vs. low-entropy contexts in the models was unclear; in particular, if separate TRFs were constructed for these groups of contexts (line 260), a single model (in which entropy was included as a factor) would seem preferable (this was suggested by the model in line 275). Please clarify the model construction and results that were obtained and which underlie the main conclusions (including how words were divided into high vs. low entropy).

3) The conclusions about listener responses to understood (Dutch) vs. not-understood (French) speech need to be tempered to account for the acoustic differences across language (and possibly across speaker – it was unclear whether the same talker was used for both languages). These points should be explicitly considered and clarified.

*Reviewer #1 (Recommendations for the authors):*

Figure 4 was very useful and I thought should appear earlier in the paper – I would put this first, so that readers see it as that.

I could not find data or analysis scripts, which would be useful to share. There is a promise that it will be available through an institutional repository but it should be available as part of the review process, no?

*Reviewer #2 (Recommendations for the authors):*

TRF analysis. As the authors mention, the TRFs will be highly correlated for different features, which makes it problematic to infer the spatiotemporal distribution of each speech feature. Can the authors take this analysis further by assessing model accuracies as before, but this time changing the lags over which the regression is computed (e.g. in sliding time windows 0-50 ms, 25-75 ms, etc.)? This would give information on processing latencies unique to a particular speech feature, going beyond what is currently presented.

Could the authors discuss what might explain their finding of specific phoneme tracking independent of acoustics, which is contra the findings of Daube et al.

I am unclear about linear mixed models presented for the 'Tracking performance of linguistic features' section (some of the issues described here apply elsewhere too). The text states that forward difference coding was used to assess the difference between consecutive models. Which p-values correspond to these contrasts? Are they reported in the tables? But these look like the coefficients of a model, not model comparisons. Significance braces are shown in the figures but the captions do not indicate what these represent exactly. Then in line 224 there is a single p-value attributed to the effect of model, which again I am unsure how to interpret (is the combined effect of the forward difference contrasts?). Finally source plots in Figure 1D, do they show the forward difference contrasts or do they show pairwise comparisons between e.g. acoustic model vs phoneme onsets, phoneme onsets vs phoneme entropy etc. To make this section clearer, in addition to providing additional detail (e.g. in the figure captions), I would consider using more familiar pairwise comparisons instead of forward difference contrasts.

Figure 2, please fully label all plots. e.g. 2A the dark vs light green bars are not distinguished (same issue for TRF plots). Do they represent the left vs right hemispheres? If so, where are the data for the high vs low entropy comparison? There is an inset panel but I think it would be clearer to show all the 'conditions' in the main graphs i.e. high low entropy, left right hemisphere. As with Figure 1, caption needs more detail e.g. what are the significance braces indicating?

Please specify how words were grouped into high and low entropy. Median split?

*Reviewer #3 (Recommendations for the authors):*

Regarding point (2) of the public review, my suggestion would be to analyze the distinction by including both low and high entropy predictor sets in a single TRF model. This way, all words' responses would be accounted for, and overlapping responses from different words would be properly controlled for. Contributions from specific sets of predictors could still be assessed by estimating models without those predictors.

Regarding point (3), a stronger case could be made, for example, with a group of French speakers listening to the same stimuli – to make comprehension orthogonal to acoustic properties.

Errors/inconsistencies

Line 531 implies that acoustic edges were modeled in 8 bands ("8-band acoustic onset spectrogram") but Figure 4 only shows a single time series.

In Figure 4A, it looks to me like the spectrogram is shifted relative to the sound wave.

In Figure 4A, the third word has lower Word Entropy than the fourth, but is assigned to the high entropy condition, while the fourth is in the low entropy condition.

The Discussion says about the initial model (before adding entropy) "Phoneme surprisal, phoneme entropy, and word frequency reconstruction accuracies were left-lateralized" (348), but I don't think I saw a statistical test of lateralization. In fact, in the model with entropy the effect of Hemisphere for phoneme features was n.s. (281).

"To isolate the effect of sentence and discourse context and control for the frequency effect we also added word frequencies in our full model" (374): I don't believe this is doing what is intended (/described). The main question asked statistically is whether word entropy modulates other predictors' TRFs. Adding a word frequency predictor will control for responses correlated with word frequency, but it will not control for other predictors' TRF shapes being modulated by word frequency.

---

## [Author Response]

Essential revisions:1) The aspects of the current results that are consistent with vs. different to prior arguments should be better clarified. As it stands, the theoretical advance of the current paper relative to much current thinking in the field is not always clear.

No previous studies compared comprehended and uncomprehended speech in a naturalistic context – this is a crucial comparison, in our view, in order to best estimate how much the brain response is driven purely by the acoustic dynamics and physical statistical properties of a natural speech stimulus. Estimating this allows us to then determine the neural responses that are associated with language comprehension over and above the demands on the brain by speech processing.

Furthermore, previous studies did not estimate the contribution of word entropy on the encoding of both acoustic (i.e., spectrogram and acoustic edges) AND linguistic lower-level features (i.e., phoneme entropy). It has of course previously been shown that word entropy explains neural response variance, but prior to our study the dynamic and interactive relationship between word-level features and low-level features was hypothesized, but not directly shown. Thus we can show that word entropy affects the encoding of both acoustic and linguistic features, which has not been shown before. We discuss later in this response letter and in the manuscript more specifically how our findings are different from particular studies in the literature. In short, our study uses a naturalistic spoken stimulus (audiobook) and compares the effect of word entropy on both acoustic and linguistic features, whereas previous research only looked at acoustic features in highly controlled experimental settings.

Because of these two key manipulations – manipulating language comprehension apart from speech processing, and directly comparing acoustic and linguistic feature encoding as a function of word entropy – we show for the first time a dynamic tradeoff between acoustic and linguistic features, which previous work focused on separately, and which was not addressed within the same study or analysis. Because we did address these issues all together, we could show that (1) Linguistic features are encoded more strongly during language comprehension than when comprehension is absent, and (2) that high word entropy enhances the encoding of lower-level acoustic and linguistic features while low word entropy suppresses it.

2) The treatment of the high-entropy vs. low-entropy contexts in the models was unclear; in particular, if separate TRFs were constructed for these groups of contexts (line 260), a single model (in which entropy was included as a factor) would seem preferable (this was suggested by the model in line 275). Please clarify the model construction and results that were obtained and which underlie the main conclusions (including how words were divided into high vs. low entropy).

We carried out a new analysis, followed the suggestions of the Reviewers, and evaluated them in a single model, and we find the same effects. The effect size was smaller than when evaluated in separate models, but the overall pattern was robust.

3) The conclusions about listener responses to understood (Dutch) vs. not-understood (French) speech need to be tempered to account for the acoustic differences across language (and possibly across speaker – it was unclear whether the same talker was used for both languages). These points should be explicitly considered and clarified.

We appreciate the Reviewer concerns here and want to explain better why acoustic differences are already accounted for in our model comparison approach. We are not comparing acoustic models directly – if we were, we agree with the Reviewer that this would be problematic. However, we are comparing how much adding linguistic features to base models that have already taken into account acoustic differences results in more variance explained in the neural response. By comparing the addition of phoneme-level and word-level features to a base model that already takes into account acoustic differences, we can assess the differences between language over and above any acoustic differences. That said, we carried out additional analyses to control for speaker identity and gender differences (detailed below), and we found the same effects; no differences compared to our previously reported findings.

Reviewer #1 (Recommendations for the authors):Figure 4 was very useful and I thought should appear earlier in the paper – I would put this first, so that readers see it as that.

We thank the reviewer for this suggestion. We agree that it would be helpful to put this figure first and we moved that figure to introduction section.

I could not find data or analysis scripts, which would be useful to share. There is a promise that it will be available through an institutional repository but it should be available as part of the review process, no?

Analysis script is shared in this link. https://github.com/tezcanf/Scripts_for_publication.git

Reviewer #2 (Recommendations for the authors):TRF analysis. As the authors mention, the TRFs will be highly correlated for different features, which makes it problematic to infer the spatiotemporal distribution of each speech feature. Can the authors take this analysis further by assessing model accuracies as before, but this time changing the lags over which the regression is computed (e.g. in sliding time windows 0-50 ms, 25-75 ms, etc.)? This would give information on processing latencies unique to a particular speech feature, going beyond what is currently presented.

We thank reviewers for this suggestion. We have now run new models with changing time lags and computed the model accuracy improvement by each feature by subtracting the reconstruction accuracy of a model from the previous model which doesn’t have the feature of interest as it was done in the first analysis where we showed the additional contribution of each feature to the reconstruction accuracy. Due to long computation time of boosting algorithm we used to compute the TRFs, we could use 17 different time lags separated by 50 ms with a 100 ms sliding window between -100 and 800 ms. We then compared the averaged reconstruction accuracies over whole brain of high and low word entropy conditions of each language (French and Dutch) with a cluster-level permutation test across time with 8000 permutations. Below graphs shows the reconstruction accuracy improvement by each feature for each time window. On x axis, accuracy of each time windows is shown on their center time. (For example, for the window between -100 ms and 0 ms, it’s shown on t = 0.05 seconds). Red bar below shows the time intervals when there was a significant interaction between language and word entropy. Analysis results shows that word entropy modulated the encoding of phoneme onset more in the comprehended language between 200 – 750 ms in LH and RH, between 150 and 650 ms in LH and between 0 and 600 ms in RH for phoneme surprisal, and it was between 550 ms and 650 ms in LH for phoneme entropy.

Could the authors discuss what might explain their finding of specific phoneme tracking independent of acoustics, which is contra the findings of Daube et al.

We extended the discussion about the encoding of phonemic features beyond acoustic features in the discussion as suggested by the reviewer.

“An interaction between language and encoding model showed that reconstruction accuracy improvement from the addition of phoneme-level features and word frequency was greater in Dutch stories, suggesting that when language is comprehended, internally-generated linguistic units, e.g., phonemes, are tracked over and above acoustic edges (Meyer, Sun, and Martin, 2020). Our findings contradict the results of Daube et al. (2019), who used articulatory features and phoneme onsets as phonetic features and showed that phoneme-locked responses can be explained by the encoding of acoustic features. It is possible that information theoretic metrics, such as surprisal and entropy, are better suited to capture the transformation of acoustic features into abstract linguistic units.”

I am unclear about linear mixed models presented for the 'Tracking performance of linguistic features' section (some of the issues described here apply elsewhere too). The text states that forward difference coding was used to assess the difference between consecutive models. Which p-values correspond to these contrasts? Are they reported in the tables? But these look like the coefficients of a model, not model comparisons.

We thank the reviewer for pointing out this confusion. We noticed that using the name for model for linear mixed effect models (LMMs) and also as the fixed effect in LMMs might be confusing, so we replaced the model with LMM for linear mixed effect models In Table 1,2 and 3. p values correspond to the contrasts Phoneme Onset – Acoustic, Phoneme Surprisal – Phoneme Onset, Phoneme Entropy – Phoneme Surprisal, Word Frequency – Phoneme Entropy. Names of the contrasts were revised in the manuscript too. We also made a mistake in the name of the contrast. It should be backward difference instead of forward. We revised it in the manuscript too. We used backward difference coding because we want to compare the accuracy improvement by each additional features on top of the previous model. For example, when we contrast Phoneme Onset – Acoustic models, Phoneme Onset model has both acoustic features (spectrogram and acoustic edges) and phoneme onset feature whereas Acoustic model only has acoustic features, so this contrast gives us the model accuracy improvement by phoneme onset feature. Revised tables are as below.

Significance braces are shown in the figures but the captions do not indicate what these represent exactly.

Figure caption is revised as below.

“Figure 2. A) Accuracy improvement (averaged over the sources in whole brain) by each feature for Dutch Stories B) Accuracy improvement (averaged over the sources in whole brain) by each feature for French Stories Braces in Figure A and B shows the significance values of the contrasts (difference between consecutive models, **** <0.0001, *** <0.001, **<0.01, * < 0.05) in linear mixed effect models (Table 2 and 3)”

Then in line 224 there is a single p-value attributed to the effect of model, which again I am unsure how to interpret (is the combined effect of the forward difference contrasts?).

P values in the text starting in line 224 shows the significance test for LMM comparison. To evaluate whether adding language and models and their interaction as fixed effects increased predictive accuracy of LMM, we compared LMMs with and without these effects using R’s anova() function. Formula for each LMM also added to the text. LMM comparison showed that adding each fixed effect and their interaction increased the predictive power and also LMM with both fixed effect and the interaction has the lowest Bayesian Information Criteria. After we made sure that adding these fixed effects increased the predictive power of the LMM, we presented the results of the LMM with these fixed effects in Table 1,2,3,4 and 5. We also added the formulas of compared LMMs in the text.

To evaluate whether adding language and models and their interaction as fixed effect increased predictive accuracy, we compared LMMs with and without these effects using R’s anova() function.

The formulas used for the LMMs were then:

LMM1: Accuracy ∼ Language∗ Models + (1+Language|subject)

LMM2: Accuracy ∼ Language+ Models + (1+Language|subject)

LMM3: Accuracy ∼ Models + (1|subject)

LMM4: Accuracy ∼ 1+ (1|subject)

LMM comparison showed that Models (LMM3 – LMM4 Δχ^2^ = 82.79, p<0.0001, LLM3 Bayesian Information Criterion (BIC): -3633.1, LMM4 BIC: -2787.5 ) , Language (LMMl2 – LMM3 Δχ^2^ = 862.02, p<0.0001, LLM2 BIC: -3693.9), and their interaction (LMM1 – LMM2 Δχ^2^ = 71.63, p <0.0001, LLM1 BIC: -3743.7) predicted the averaged reconstruction accuracies.

Finally source plots in Figure 1D, do they show the forward difference contrasts or do they show pairwise comparisons between e.g. acoustic model vs phoneme onsets, phoneme onsets vs phoneme entropy etc. To make this section clearer, in addition to providing additional detail (e.g. in the figure captions), I would consider using more familiar pairwise comparisons instead of forward difference contrasts.

We thank the reviewer for expressing this preference and helping us be more consistent. We used a backward difference contrast in our analysis. Source plots in Figure 2D (Figure 1D in the previous version of the manuscript) shows the backward difference contrasts which corresponds to the sources where a specific linguistic feature incrementally increased the model accuracy compared to previous model. We used a backward difference contrast because we wanted to investigate if adding each acoustic or linguistic feature increased the model accuracy compared to the previous model which doesn’t have that particular feature. We added a more detailed description to the figure caption to make it clearer.

“Figure 2. (D) Source points where reconstruction accuracies of the model were significantly different than previous model. Accuracy values shows how much each speech feature increased the reconstruction accuracy compared to the previous model.”

Figure 2, please fully label all plots. e.g. 2A the dark vs light green bars are not distinguished (same issue for TRF plots). Do they represent the left vs right hemispheres? If so, where are the data for the high vs low entropy comparison? There is an inset panel but I think it would be clearer to show all the 'conditions' in the main graphs i.e. high low entropy, left right hemisphere. As with Figure 1, caption needs more detail e.g. what are the significance braces indicating?

We apologize for the confusion. Dark and light color bars represent high and low word entropy condition. Braces indicate the significant different between high and low entropy word conditions. In Figure 3 A (Figure 2A in the previous version of the manuscript), reconstruction accuracies are averaged over whole brain to show the interaction between word entropy and language. All figures are labelled and figure caption is updated in the revised version.

Please specify how words were grouped into high and low entropy. Median split?

Instead of splitting them with the median value, we picked the entropy value that divided the auditory signal into equal lengths to have equal length of zero values for predictors. Words in each story part were divided into two conditions; high and low word entropy, so that each condition has equal length of signal not to have a bias on one condition due to the amount of training data.

Reviewer #3 (Recommendations for the authors):Regarding point (2) of the public review, my suggestion would be to analyze the distinction by including both low and high entropy predictor sets in a single TRF model. This way, all words' responses would be accounted for, and overlapping responses from different words would be properly controlled for. Contributions from specific sets of predictors could still be assessed by estimating models without those predictors.

We thank the reviewer for pointing this confusion. We didn’t bin the words as they were in separate trials. After generation of the acoustic and phonetic features for continuous stimuli, we projected predictors into a higher dimensional space by multiplying them with a weight matrix that zeros out all the predictors of high entropy words in low entropy word condition and vice versa. However, MEG signal was intact. This creates sparser predictors for each condition to create a contrast between two conditions to account for the nonlinear changes in the response function (viz., it is also used for amplitude binning by Drennan and Lalor, 2019). We fitted 2 separate models for each condition to compare the accuracies of conditions. As you also mentioned, in separate models, when a model is predicting the response for high entropy words, it is also predicting the absence of a response in low entropy words which gives us the opportunity to contrast the effect word entropy between conditions.

Following your suggestion, we also fitted one single model for those predictors and predicted the reconstruction accuracies of high and low entropy word conditions by subtracting accuracies of models without those predictors from the full model which has all predictors. For example, to calculate the accuracy of phoneme features in low entropy word condition we used below method.

Full_model = [spectrogram_low, spectrogram_high, acoustic_edge_low, acoustic_edge_high, phoneme_onset_low, phoneme_onset_high, phoneme_surprisal_low, phoneme_surprisal_high, phoneme_entropy_low, phoneme_entropy_high, word_freq_low, word_freq_high]

Full_model_minus_low_word_entropy_phonemes = [spectrogram_low, spectrogram_high, acoustic_edge_low, acoustic_edge_high, phoneme_onset_high, phoneme_surprisal_high, phoneme_entropy_high, word_freq_low, word_freq_high]

R^2^ of phonetic features in low entropy word condition = R^2^ of Full_model – R^2^ of Full_model_minus_low_entropy_phonemes

With this method, we calculated the accuracies of phoneme features in low entropy condition, phoneme features in high entropy condition, acoustic features in low entropy condition and acoustic features in high entropy condition.

To compare the reconstruction accuracies of each feature in each condition, we fitted a linear mixed effect model with the below formula to analyze the interaction between word entropy and language for acoustic and phonemic features.Accuracy ∼ Language + Word Entropy + Hemisphere + Word Entropy ∗ Language + (1+ Language + Hemisphere |Subject)

**Author response table 1. sa2table1:** LMM results of reconstruction accuracies for Phoneme Features.

	Estimate	Std. Error	df	t value	Pr(>|t|)	
(Intercept)	3.29E-05	5.45E-06	43.07	6.03	3.34E-07	****
French	-2.48E-05	4.06E-06	164.00	-6.12	6.75E-09	****
Low Word Entropy	-2.00E-05	4.06E-06	164.00	-4.92	2.10E-06	****
Right Hemisphere	-3.73E-06	2.87E-06	164.00	-1.30	1.95E-01	
French: Low Word Entropy	1.09E-05	5.74E-06	164.00	1.90	5.96E-02	.

**Author response table 2. sa2table2:** LMM results of reconstruction accuracies for Acoustic Edges.

	Estimate	Std. Error	df	t value	Pr(>|t|)	
(Intercept)	4.01E-04	3.82E-05	32.41	10.50	5.85E-12	****
French	1.48E-05	2.16E-05	140.00	0.69	4.93E-01	
Low Word Entropy	-5.78E-06	2.64E-05	140.00	-0.22	8.27E-01	
Right Hemisphere	9.76E-05	4.96E-05	28.02	1.97	5.91E-02	.
French: Low Word Entropy	-5.88E-05	3.05E-05	140.00	-1.93	5.60E-02	.

**** <0.0001, *** <0.001, ** <0.01, * < 0.05,. <0.1

We found that interaction between language and word entropy both for acoustic and phonemic features which were marginally significant (p=0.0596 for phonemic features and p=0.056 for acoustic features). So, this effect was stronger when we fitted 2 separate model that predicts the response only for that condition.

Author response image 1 shows the opposite interaction between language and word entropy for acoustic and phonemic features. We also compared the TRF weights of each condition in the full model. Results are very similar to the TRF weights obtained by two separate models shown in Figure 3 A and B in the manuscript.

**Author response image 1. sa2fig1:** Full Model. Light orange and light green represent Low Word Entropy condition, dark orange and dark green represent High Word Entropy condition for Dutch and French stories, respectively. (Braces indicate the significant different between high and low entropy word conditions, **** <0.0001, *** <0.001, ** <0.01, * < 0.05 ) (A) Reconstruction accuracy interaction between word entropy and language for acoustic features (B) Reconstruction accuracy interaction between word entropy and language for phoneme features (C-D) Acoustic Edge TRFs on LH and RH (E-F) Phoneme Features TRFs on LH and RH (G) Sources where the main effect of Language and Word Entropy, and interaction are found.

Regarding point (3), a stronger case could be made, for example, with a group of French speakers listening to the same stimuli – to make comprehension orthogonal to acoustic properties.

We thank the reviewer for this comment. As we have already included acoustic features as a base model and compared how much phonetic features increased the model accuracy relative to the base acoustic model rather than directly comparing the accuracies of acoustic models for our first question, and we investigated how sentence context modulated acoustic and phonetic features for our second and third question, these interaction effects are independent from the particular acoustic features which may be related to acoustic differences in stimuli. We agree with the reviewer that if we were only comparing the model accuracies of acoustic features, then acoustic differences related to speaker, language, or gender differences would be a confound in the model accuracy differences. We appreciate the Reviewer’s suggestion that comparing French speakers to Dutch speakers listening to our stimuli might shed light on the problem, however, comparing two different group of participants (Dutch and French speakers) listening to the same Dutch stimuli would, rather than ameliorate the concern, lead to unsurmountable challenges in comparison of model accuracies between groups of participants with different language experience. In this case individual differences in signal-to-ratio of neural signal would be problematic when comparing the model accuracies of two groups, which are likely to be greater than the contribution of the any acoustic difference which are already modelled in the base models (Crosse et al., 2021).

We added a new table (Table 10) that shows which story part was read by different speakers in the revised manuscript. French stories were read by both a woman and a man speaker, however Dutch stories were read by women speakers only. In the first version of our manuscript, we compared the first 4 parts of Dutch stories read by the same woman with the 4 parts of French stories read by a woman and a man speaker in order to have similar amount of training data for both languages because the amount of training data is also factor for the model accuracy. Then we repeated the same analysis with the second group of four Dutch story parts that read by another woman speaker, and crucially, we find the same effects (see reported results in Figure 3 and Figure 4 in the manuscript). To test if gender difference has an impact on the results of our first question, we repeated the same analysis that presented in Table 1 and compared the first 2 part of Dutch stories (read by women 1 in Table 10 in the revised manuscript) with the first 2 parts of French stories (read by women 3). Then compared the first 2 parts of Dutch stories (read by women 1) with second 2 parts of French stories (read by man 1) Linear mixed effect model results are presented below. For each comparison we found the same interaction between languages and model accuracy improvement by phoneme level features.

**Author response table 3. sa2table3:** LMM results of reconstruction accuracies for Dutch (first 2 parts) and French (first 2 parts) stories.

	Estimate	Std. Error	t value	Pr(>|t|)	
(Intercept)	2.92E-03	2.72E-04	10.72	1.99E-10	***
Language (French – Dutch)	-6.44E-04	1.52E-04	-4.23	3.13E-04	***
Phon. Onset – Acoustic	5.57E-05	1.23E-05	4.52	1.13E-05	***
Phon. Surprisal – Phon. Onset	8.10E-05	1.23E-05	6.57	5.04E-10	***
Phon. Entropy – Phon. Surprisal	1.25E-04	1.23E-05	10.10	2.00E-16	***
Word Frequency – Phon. Entropy	1.22E-04	1.23E-05	9.88	2.00E-16	***
Language: Phon. Onset – Acoustic	-4.17E-05	1.74E-05	-2.39	1.80E-02	*
Language: Phon. Surprisal – Phon. Onset	-6.73E-05	1.74E-05	-3.86	1.58E-04	***
Language: Phon. Entropy – Phon. Surprisal	-1.06E-04	1.74E-05	-6.09	6.30E-09	***
Language:Word Frequency – Phon. Entropy	-1.27E-04	1.74E-05	-7.29	8.80E-12	***

**Author response table 4. sa2table4:** LMM results of reconstruction accuracies for Dutch (first 2 parts) and French (second 2 parts) stories.

	Estimate	Std. Error	t value	Pr(>|t|)	
(Intercept)	2.92E-03	2.72E-04	10.72	2.00E-10	***
Language (French- Dutch)	-1.09E-03	1.75E-04	-6.20	2.45E-06	***
Phon. Onset – Acoustic	5.57E-05	1.18E-05	4.71	4.79E-06	***
Phon. Surprisal – Phon. Onset	8.10E-05	1.18E-05	6.86	1.03E-10	***
Phon. Entropy – Phon. Surprisal	1.25E-04	1.18E-05	10.55	2.00E-16	***
Word Frequency – Phon. Entropy	1.22E-04	1.18E-05	10.32	2.00E-16	***
Language: Phon. Onset – Acoustic	-4.38E-05	1.67E-05	-2.62	9.49E-03	**
Language: Phon. Surprisal – Phon. Onset	-7.34E-05	1.67E-05	-4.40	1.87E-05	***
Language: Phon. Entropy – Phon. Surprisal	-1.17E-04	1.67E-05	-7.02	4.25E-11	***
Language:Word Frequency – Phon. Entropy	-1.39E-04	1.67E-05	-8.32	1.91E-14	***

Furthermore, to test if gender difference has an impact on the results of second and third questions, we repeated the same analysis that presented in Figure 3 and 4, and Table 6,7,8 and 9 in the manuscript for the story parts mentioned above. Below tables shows the results of LMMs. Similar to the previous results in the manuscript we found an opposite interaction between language and word entropy for acoustic and phoneme features. These results suggest that gender difference of the speakers doesn’t have an effect on the results.

**Author response table 5. sa2table5:** LMM results of reconstruction accuracies for Phoneme Features (First 2 parts of Dutch stories and first 2 parts of French stories).

	Estimate	Std. Error	df	t value	Pr(>|t|)	
(Intercept)	4.92E-05	7.45E-06	56.11	6.60	1.56E-08	***
French	-5.92E-05	6.46E-06	164.00	-9.17	2.00E-16	***
Low Word Entropy	-4.89E-05	6.46E-06	164.00	-7.57	2.62E-12	***
Right Hemisphere	-3.68E-06	4.57E-06	164.00	-0.81	4.21E-01	
French: Low Word Entropy	4.13E-05	9.13E-06	164.00	4.52	1.19E-05	***

**Author response table 6. sa2table6:** LMM results of reconstruction accuracies for Acoustic Edges (First 2 parts of Dutch stories and first 2 parts of French stories).

	Estimate	Std. Error	df	t value	Pr(>|t|)	
(Intercept)	1.45E-04	3.40E-05	45.16	4.27	1.00E-04	***
French	-2.16E-06	2.61E-05	164.00	-0.08	9.34E-01	
Low Word Entropy	3.58E-05	2.61E-05	164.00	1.37	1.72E-01	
Right Hemisphere	6.05E-05	1.85E-05	164.00	3.28	1.27E-03	**
French: Low Word Entropy	-1.20E-04	3.69E-05	164.00	-3.24	1.43E-03	**

**** <0.0001, *** <0.001, ** <0.01, * < 0.05

**Author response table 7. sa2table7:** LMM results of reconstruction accuracies for Phoneme Features (First 2 parts of Dutch stories and second 2 parts of French stories).

	Estimate	Std. Error	df	t value	Pr(>|t|)	
(Intercept)	4.97E-05	6.73E-06	69.14	7.39	2.55E-10	***
French	-6.57E-05	6.37E-06	164.00	-10.31	2.00E-16	***
Low Word Entropy	-4.89E-05	6.37E-06	164.00	-7.67	1.45E-12	***
Right Hemisphere	-4.77E-06	4.51E-06	164.00	-1.06	2.91E-01	
French: Low Word Entropy	3.78E-05	9.01E-06	164.00	4.20	4.38E-05	***

**Author response table 8. sa2table8:** LMM results of reconstruction accuracies for Acoustic Edges (First 2 parts of Dutch stories and second 2 parts of French stories).

	Estimate	Std. Error	df	t value	Pr(>|t|)	
(Intercept)	1.41E-04	3.06E-05	47.69	4.60	3.11E-05	***
French	-4.81E-06	2.43E-05	164.00	-0.20	8.44E-01	
Low Word Entropy	3.58E-05	2.43E-05	164.00	1.47	1.43E-01	
Right Hemisphere	6.87E-05	1.72E-05	164.00	4.00	9.69E-05	***
French: Low Word Entropy	-1.33E-04	3.44E-05	164.00	-3.87	1.55E-04	***

**** <0.0001, *** <0.001, ** <0.01, * < 0.05

Errors/inconsistenciesLine 531 implies that acoustic edges were modeled in 8 bands ("8-band acoustic onset spectrogram") but Figure 4 only shows a single time series.

In the figure we took the average of 8-bands for acoustic edges. We revised it now to show 8-bands separately.

In Figure 4A, it looks to me like the spectrogram is shifted relative to the sound wave.

Yes, it was indeed shifted. We corrected it.

In Figure 4A, the third word has lower Word Entropy than the fourth, but is assigned to the high entropy condition, while the fourth is in the low entropy condition.

Coloring was done manually, so it was a mistake. We corrected it in the revised figure 1.

The Discussion says about the initial model (before adding entropy) "Phoneme surprisal, phoneme entropy, and word frequency reconstruction accuracies were left-lateralized" (348), but I don't think I saw a statistical test of lateralization. In fact, in the model with entropy the effect of Hemisphere for phoneme features was n.s. (281).

We apologize for this mistake. The statistical analysis that we didn’t find any lateralization was the effect of word entropy on the encoding of acoustic features. There is a significant lateralization effect of the accuracy improvement by each linguistic feature with Dutch stories but not with French stories. Unfortunately, we forgot to add that to our manuscript before. It’s added to the Results section as below and we also corrected the discussion.

Results:

“Figure 2-D shows the sources where each feature incrementally increased the reconstruction accuracy compared to previous model. We fitted a liner mixed effect model to test the accuracy improvement by each linguistic feature for lateralization by taking the average of the contrasts shown in Figure 2-D for each hemisphere. We used the below formulas for LMM.

LMM1: Accuracy ∼ Hemisphere + (1|subject)

LMM2: Accuracy ∼ 1+ (1|subject)

We compared LMMs with and without Hemisphere effect using R’s anova() function. LMM comparison showed that Hemisphere (LMM1 – LMM2 Δχ^2^ = 4.03 , p<0.05, LLM1 Bayesian Information Criterion (BIC): -3331.5, LMM2 BIC: -3329.5 ) predicted the averaged reconstruction accuracies in Dutch stories but not in French stories. We reported the results of LMM1 in Table 4 and Table 5.”

In Dutch stories, linguistic features increased the reconstruction accuracy mostly on the left hemisphere however in French stories only Phoneme Onset and Entropy slightly increased the reconstruction accuracies and we couldn’t find any significant lateralization effect.

Discussion:

“Comparison of the reconstruction accuracies on the source level showed that, for Dutch stories, accuracy improvement by the linguistic features were left lateralized. However, for uncomprehended French stories, we couldn’t find a significant lateralization effect.”

"To isolate the effect of sentence and discourse context and control for the frequency effect we also added word frequencies in our full model" (374): I don't believe this is doing what is intended (/described). The main question asked statistically is whether word entropy modulates other predictors' TRFs. Adding a word frequency predictor will control for responses correlated with word frequency, but it will not control for other predictors' TRF shapes being modulated by word frequency.

We are thankful to the reviewer for pointing out this issue. We agree that adding word frequency as a predictor doesn’t control for the modulation of other predictors by word frequency. As we also mentioned in the Discussion, we still see the modulation of acoustic features in French stories, however in uncomprehended language we don’t expect to see the effect of context, so this modulation should be driven by the recognition of frequent words. We revised that part as following. To control for the responses correlated with word frequency we also added word frequencies in our full model and compared the reconstruction accuracy improvement explained by acoustic and phoneme features.